

# Large simulated radiative effects of smoke in the south-east Atlantic

Hamish Gordon[1], Paul R. Field[1,2], Steven J. Abel[2], Ben T. Johnson[2], Mohit Dalvi[2], Daniel P. Grosvenor[1], Adrian A. Hill[2], Annette K. Miltenberger[1], Masaru Yoshioka[1], and Ken S. Carslaw[1]

[1]School of Earth and Environment, University of Leeds, LS2 9JT, United Kingdom
[2]Met Office, Fitzroy Road, Exeter, EX1 3PB, United Kingdom

*Correspondence to:* Hamish Gordon hamish.gordon@cern.ch

**Abstract.** A $1200\,\mathrm{km}$-square area of the tropical south Atlantic Ocean near Ascension Island is studied with the HadGEM climate model at convection-permitting and global resolutions for a ten-day case study period in August 2016. During the simulation period, a plume of biomass burning smoke from Africa moves into the area and mixes into the clouds. We examine the interaction of the smoke with clouds and find it has substantial instantaneous direct, indirect and semi-direct radiative
effects, which vary in magnitude between model configurations.

The region of interest is simulated at $4\,\mathrm{km}$ resolution, with no parameterised convection scheme. The simulations are driven by, and compared to, the HadGEM global model, running at approximately $65\,\mathrm{km}$ resolution. For the first time, the UK Chemistry and Aerosol model UKCA is included in a regional model with prognostic aerosol number concentrations advecting in from the global model at the boundaries of the region.

The smoke aerosol is simulated realistically, and is found to affect dynamical, microphysical and radiative properties of the atmosphere across the region. The model captures the large-scale horizontal transport of the aerosol adequately, approximately reproducing a transition from pristine to polluted conditions. However, for some of the simulation, the smoke is around 1km too low in altitude and therefore mixes into the clouds earlier than observed. Fire emissions increase the total aerosol burden by a factor 3.7 and cloud droplet number concentrations by a factor of 3, which is consistent with MODIS observations. Strong
localised perturbations to heating and cooling rates due to the smoke affect the dynamics: in the regional model, the inversion height is reduced by up to $200\,\mathrm{m}$ when smoke is included. The smoke also affects precipitation, to an extent which depends on the model microphysics. The microphysical and dynamical changes lead to an increase in liquid water path of $60\,\mathrm{g\,m^{-2}}$ relative to a simulation without smoke aerosol, when averaged over the polluted period. This increase is uncertain, and smaller in the global model. It is mostly due to radiatively driven dynamical changes: the reduced entrainment of dry air from above
the cloud layer, rather than precipitation suppression by aerosol.

The smoke has substantial direct radiative effects of $+11.4\,\mathrm{W\,m^{-2}}$ in the regional model, when averaged over the polluted five days of our case study. The semi-direct radiative effect of the smoke, $-30.5\,\mathrm{W\,m^{-2}}$, is larger than the indirect radiative effect, $-10.1\,\mathrm{W\,m^{-2}}$. However, the radiative effects are sensitive to the model set-up: the semi-direct effect is smaller in the global model, and also in a simulation with the Kogan (2013) parameterisation of autoconversion and accretion instead of the
default, from Khairoutdinov & Kogan (2002). Furthermore, we simulate a liquid water path that is biased high compared to satellite observations by 22% on average, and this leads to high estimates of the domain-averaged aerosol direct effect and the effect of the aerosol on cloud albedo. With these caveats, we simulate a large net cooling across the region, of $-27.6\,\mathrm{W\,m^{-2}}$.





# 1 Introduction

Marine boundary layer clouds are a substantial source of uncertainty in climate models (Bony and Dufresne, 2005; Wood, 2012; Schneider et al., 2017). In the tropical Atlantic Ocean south of the Equator, a broad region of subsidence leads to one of Earth's largest stratocumulus decks, off the coast of Africa. During the biomass burning season, especially in August and September,

emissions from the African plateau form a layer of smoke. Depending on meteorological conditions, the smoke can advect out over the marine boundary layer, and above these clouds, at least as far as Ascension Island (Haywood et al., 2003; Adebiyi et al., 2015; Das et al., 2017). When above clouds, the absorbing smoke layer leads to a direct and a semi-direct radiative effect (Hansen et al., 1997; Johnson et al., 2004; Wilcox, 2010). The smoke layer slowly subsides and meets the gradually deepening marine boundary layer in the neighbourhood of Ascension Island and St Helena (Adebiyi et al., 2015). Generally,

the smoke is entrained into the clouds during, or before, the transition from stratocumulus to trade cumulus. This transition is driven by increasing sea surface temperatures (e.g. Sandu and Stevens, 2011) and can be modulated both by entrainment of dry air (e.g. Wyant et al., 1997) and by precipitation (Yamaguchi et al., 2017). Once entrained, smoke in the boundary layer can have very different effects to smoke aloft (Hill et al., 2008; Koch and Del Genio, 2010). The region of mixing and transition, shown schematically in Figure 1, covers huge swathes of ocean and may be very sensitive to anthropogenic aerosols. As

indicated in the figure, we hypothesise that indirect or semi-direct radiative effects dominate the direct effect of smoke aerosols west of about 5°W. However, many climate models respond differently to smoke aerosols (Das et al., 2017) and have difficulty representing smoke episodes in this region (Peers et al., 2016). Therefore the area is a focus of current fieldwork and simulation activity.

    Long-term field observations both in Africa and over the ocean are still sparse. To start to remedy these difficulties, four

major campaigns in 2016-7 may yield greatly improved understanding of this complex environment (Zuidema et al., 2016). The LASIC (Layered Smoke Interacting with Clouds) deployment of the Atmospheric Radiation Measurement (ARM) mobile research facility to Ascension Island over 2016 and 2017 has provided sorely needed long term measurements in a particularly critical location. The NASA Observations of Aerosols above Clouds and their Interactions (ORACLES) flights from Walvis Bay (Namibia) in 2016 and Sao Tomé in 2017 have characterised the aerosol-radiation interactions close to the African coast and, on

several flights, further out to the mid-Atlantic area. The AErosol RadiatiOn and CLOuds in Southern Africa (AEROCLO-sA) aircraft measurements and field station in Namibia cover a similar region. Finally the CLouds and Aerosol Radiative Impacts and Forcing (CLARIFY) aircraft campaign from Ascension Island has focused on the region where the aerosols and clouds mix directly.

    Modelling the stratocumulus-to-cumulus transition (SCT) is challenging because the structure of the aerosols and clouds is

often complex, with multiple thin layers of aerosol and cloud which may or may not be coupled together. This also complicates the interpretation of satellite retrievals commonly used in model evaluation (Haywood et al., 2004). Many earlier modelling studies focused on the semi-direct effects of smoke on clouds (e.g. Johnson et al., 2004; Sakaeda et al., 2011), and regional modelling of marine boundary layer cloud fields (e.g. Wang et al., 1993; Sandu and Stevens, 2011) as well as general circulation modelling (Randles and Ramaswamy, 2010; Sakaeda et al., 2011; Das et al., 2017). More recently, large eddy simulations of the





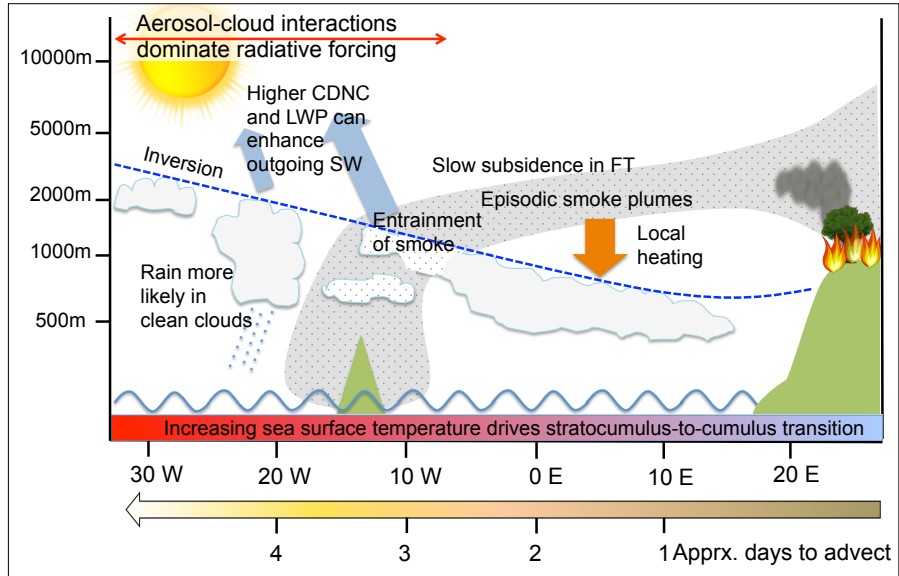

**Figure 1.** Schematic showing the large-scale situation in the south-east Atlantic, approximately along the pathway of the smoke aerosol. The green triangular island represents Ascension Island. We focus here on a case in which the entrainment of smoke is simultaneous with the stratocumulus-to-cumulus transition, but smoke may also mix into the cloud deck closer to the African continent.

SCT in the presence of smoke aerosol (Yamaguchi et al., 2015; Zhou et al., 2017) have produced interestingly varied results. In brief, Yamaguchi et al. (2015) find that smoke aerosol delays the SCT and increases the liquid water path, while Zhou et al. (2017) find the reverse. The agreement of the models with observation-based analysis is similarly mixed: the study of Costantino and Bréon (2013) suggests aerosol decreases liquid water path on one hand, but enhances cloud cover, thus delaying the SCT, on the other. The correlation of absorbing aerosol optical depth with fractional cloud cover, however, was recently confirmed by (Adebiyi and Zuidema, 2018).

A noteworthy regional modelling study of the south-east Atlantic by Lu et al. (2018) using the WRF model was published as this manuscript was being prepared for submission. The authors studied a $16 \times 20°$ area of ocean; our regional domain covers a similar area to their 'remote' sub-region. They focused more on the large-scale radiative effects of smoke than did the LES studies of the SCT. The main conclusion, which we discuss in more detail later, was that the indirect radiative effect is larger, by around a factor of two in the remote region, than the sum of the direct and semi-direct radiative effects.

This study aims to establish the HadGEM climate model (Walters et al., 2017b) as a tool to represent aerosol-cloud-climate interactions at convection-permitting resolution in the tropical South Atlantic. As the HadGEM climate model is also the UK Met Office numerical weather prediction model (sometimes referred to as the 'Unified Model' or UM), it is also expected to perform well when run regionally for short periods of time at higher resolution. In this paper, we test a high-resolution configuration and see what information it can add to global model simulations. The increased resolution is expected to allow us to better represent rapidly varying cloud properties, while the driving global model provides a realistic treatment of the more





slowly varying aerosol and dynamics: aerosol emissions from continents can be propagated and processed as they move across the ocean, before they enter the regional domain. Once evaluated, the regional model can be developed to treat aerosol-cloud interactions across the stratocumulus-to-cumulus transition in more detail in subsequent studies. The first such development for the regional model we have planned is to use the aerosol from the UM to drive the CASIM two-moment cloud microphysics

scheme (Shipway and Hill, 2012; Grosvenor et al., 2017).

In Section 2, we introduce our $\sim 65\,\mathrm{km}$-resolution (N216) version of the global climate model with two-moment aerosol microphysics (prognostic mass and number concentrations). This model is used to drive a simulation of a roughly square domain of length approximately $1200\,\mathrm{km}$ (strictly $10.8°$ of latitude and longitude) surrounding Ascension Island with a grid spacing of $4\,\mathrm{km}$, intended to allow us to turn off the convection parameterisation in the model. This is the first time prognostic

aerosol number concentrations are included in a regional UM configuration with realistic meteorology, except for the idealised demonstration case study of Planche et al. (2017). Therefore we give particular emphasis to the evaluation of the cloud droplet number concentration (CDNC) in our model. We made some adjustments to the model to improve its representation of CCN and CDNC in the region of interest, and describe these in Section 3.

In Section 4, we introduce the satellite retrievals we use. In Section 5, we examine the large-scale transport of aerosol with

a trajectory analysis. In Section 6, we compare the model quantitatively to these satellite observations and to data from the ARM facility on Ascension Island and find the model and observations to be generally in agreement. Therefore, we present a comparison of our realistic simulation with a simulation in which biomass burning emissions are switched off, and examine the effect of the aerosol on temperature, precipitation and radiation balance in Sections 7, 8 and 9. We consider the sensitivity of the radiative effects we find to the cloud microphysics by changing the parameterisation of autoconversion and accretion

rates, and compare the higher-resolution regional model with the global model. We highlight some areas where further work is needed in our conclusions.

## 2 The regional climate configuration of the Unified Model with UKCA aerosol

### 2.1 Model structure and spin-up

The global model version used in this study is a hybrid of the regional configuration of the UK Met Office Unified Model,

hitherto used mostly for numerical weather prediction, and the HadGEM-UKCA configuration for climate modelling. It is based on the GA6.1 configuration (Walters et al., 2017b) of version 10.3 of the Unified Model. The horizontal resolution is around $65\,\mathrm{km}$, and there are 70 vertical levels from the surface to $70\,\mathrm{km}$ altitude, spaced such that the vertical resolution is $50\,\mathrm{m}$ at the surface and approximately $200\,\mathrm{m}$ at the level of low clouds. The model version includes the ENDGAME dynamical core (Wood et al., 2014) and the UKCA chemistry and aerosols package (Morgenstern et al., 2009; O'Connor et al., 2014). In

UKCA, prognostic aerosol number concentrations are represented with GLOMAP (Mann et al., 2010), which contains five log-normal aerosol size modes and includes sulphate, sea-salt, black carbon and organic carbon chemical components. To maintain computational efficiency while incorporating the GLOMAP-mode aerosol scheme, we use a reduced version of the UKCA chemistry scheme in which the sulphur cycle and the production of secondary organic aerosol are represented prognostically



but the oxidants OH, ozone, $HO_2$ and $H_2O_2$ are provided as monthly mean climatologies derived from a simulation with online chemistry (Walters et al., 2017a).

Roughly the same configuration is applied to the 4 km-resolution regional simulation centred at $10°$S, $12°$W, with a few differences in the parameters used, for example, in the boundary layer scheme. This regional model has a slightly higher

vertical resolution: it also has 70 vertical levels, but they are spread over the first 40 km. The model grid boxes at the boundary layer top altitude, which is quite variable but usually around 1800 m in this period, are around 200 m deep in the regional model and 300 m in the global model. The horizontal and vertical resolution is chosen to allow us to represent a large area of ocean, covering a substantial area of the stratocumulus-to-cumulus transition regime, without the simulations becoming too expensive. The major difference in the regional model compared to the global model is that the parameterised convection

scheme is switched off. At the edges of the model, a transition from the global model to the regional model is made over nine 4 km-grid boxes. The regional model cannot be expected to resolve processes at the scale of its own 4 km grid boxes in the spatial spin-up region close to its boundaries, because the inputs from the lower-resolution driving model will reduce the variability between grid boxes close to the edges of the regional domain. Therefore grid boxes within 200 km of this boundary are excluded from calculations of the regional mean results we present later.

The surface of Earth is not represented interactively in our model set-up. Sea surface temperatures (SSTs) are fixed from the OSTIA temperature record (Donlon et al., 2012) via the UM start-dumps. They are initialised from the SST on 20 July for the spin-up phase and then re-initialised at the start of the model run on 1 August from the SST on 1 August. Emissions of gases and particles from the sea and, in the global model, the land surfaces, are read in from inventories (described below) rather than calculated online, except for emissions of sea spray and DMS, which are parameterised online. There is no land in our regional

simulation, as Ascension Island ($7.9°$S, $14.3°$W) is not included in the Unified Model when the orography is reconfigured from the global 65 km resolution. While the peak of Green Mountain is 800 m high and thus some localised orographical effects are likely, the area of the island is of order $50 \, \text{km}^2$ and the domain is $1.4 \times 10^6 \, \text{km}^2$ so we assume the island has negligible effects on domain-averaged quantities.

The global model is initialised from the UM global operational meteorology and climatological aerosol fields at 0000 UTC

on 20 July 2016. It is then run for 12 days, until 2400 UTC on 31 July, to allow the biomass burning aerosol from Africa to advect out to the domain around Ascension Island. This spin-up time also allows the model to pick up more appropriate natural and anthropogenic aerosol emissions from the climatological emissions files, which include the seasonal cycle. The global model is nudged every six hours by ERA-interim wind fields above 1800 m (the $15^{th}$ model level from the surface). As we do not nudge to temperature, following the recommendations of Zhang et al. (2014), the global model is sensitive to semi-direct

radiative effects of aerosol. On 1 August at 0000 UTC, the regional model is started, and both models are then run until 2359 UTC on 10 August. The regional model is not nudged, to allow dynamical effects full freedom to manifest themselves, but it is forced at the boundaries. This leads to a slow deviation of the wind fields in the regional model from those in the global model over the 10-day simulation. This is visible in Supplementary Figure S7, but the effect is relatively small and not expected to have a large impact on regional mean results.



## 2.2 Aerosol emissions and transport

We use biomass burning emissions from the MODIS-based Fire Energetics and Emissions Research (FEER) inventory (Ichoku and Ellison, 2014) for 2016. These have daily time resolution and $0.1°$ spatial resolution. We assume the diameter of the emitted particles is $120\,\mathrm{nm}$ and the emissions are highest at the surface, then taper down to reach zero at $3\,\mathrm{km}$ above ground level. These emissions replace the monthly-mean climatological emissions (from GFED version 3) that are appropriate for the climate model. The resulting change in the mass of smoke aerosol found in our region of interest is small compared to the uncertainty in the overall biomass emitted (around 25%). The FEER inventory has higher emissions, but, in the Global Atmosphere configurations of HadGEM, emissions in the GFED inventory in the UKCA model are scaled up by a factor 2. Compared to the scaled GFED emissions, the FEER emissions are roughly 25% lower, although this also depends on time, as the FEER inventory has a higher time resolution than the version of the GFED emissions usually used in the model. In any case, the main factor that determines how much aerosol is found over the ocean is advection rather than emission (Gunnar et al.). For all other anthropogenic and natural emissions, the CMIP5 climatologies (Taylor et al., 2012) used in the standard model configurations are retained.

The biomass burning emissions are placed initially into the insoluble Aitken mode, which precludes them from activating to form cloud droplets. However, if they become coated with at least one monolayer of condensable material during any model timestep, they 'age' into the Aitken soluble mode. In our simulations, by the time the particles have advected into our regional domain, between 86% and 96% of black carbon mass is found in the soluble modes due to condensation of sulphuric acid and secondary organics, and coagulation with other particles.

## 2.3 Cloud microphysics

Aerosol activation to cloud droplets is calculated explictly in the UKCA-Activate scheme (West et al., 2014) which implements the parameterisation of Abdul-Razzak and Ghan (2000). The activation scheme in both global and regional models is calculated for a probability distribution function (PDF) of updraft velocities in each grid box centred around the large-scale vertical velocity: this means the scheme is run twenty times per grid box per timestep with an increasing updraft velocity. The number of aerosol that activate is the expectation value of the PDF of cloud droplet number concentration over the twenty bins of updraft velocity. Cloud droplet numbers are calculated diagnostically by running the activation scheme on each model timestep from scratch, without consideration of how many cloud droplets were present before. The cloud droplet number concentration is then passed to the radiation and microphysics schemes.

The cloud microphysics in our model is single-moment, in that the mass of liquid water, but not the cloud droplet number, is advected by the model and retained in memory between model timesteps. The scheme is based on Wilson and Ballard (1999) but we use improved warm rain parameterisations following Boutle et al. (2014) with their suggested modified versions to the autoconversion and accretion rates. These rates are used everywhere in the regional model, but only where the convection scheme is not triggered in the global model. In the region of interest in the global model, the convection scheme fires rarely but not so rarely that its effects can be neglected: it is responsible for around 20% of the rain.





In our model, two potentially important 'second indirect' effects of cloud droplet number concentration on radiative transfer are not simulated, because the CDNC is not treated prognostically. This means that sedimentation of cloud water is assumed to be independent of the droplet size. In reality, smaller droplets fall more slowly, which leads to increased water content at the top of clouds, and this is more susceptible to evaporation when dry air is entrained into the clouds from above (Bretherton et al., 2007; Zhou et al., 2017). Furthermore, in our simulation, evaporation of cloud droplets is also independent of their size. Smaller droplets resulting from higher aerosol concentrations tend to to evaporate more quickly. This leads to greater turbulence in the boundary layer, hence increased entrainment of dry air, and a positive feedback that further increases evaporation and reduces the liquid water path (Wang et al., 2003; Hill et al., 2008). Further studies with two-moment cloud microphysics will enable these effects to be quantified.

As neither 4 km nor global resolution can represent individual clouds, we employ the Smith (1990) scheme for sub-grid cloud in our regional model and the 'pc2' scheme (Wilson et al., 2008) in the global model. The result is a fraction of cloud coverage in each model grid-box. To calculate this fraction, the Smith (1990) scheme assumes the probability density function of liquid water mass mixing ratio is triangular and that cloud may begin to form in a grid-box when a critical relative humidity (less than 100%) is exceeded. The 'pc2' scheme, standing for 'prognostic cloud, prognostic condensate', extends the Smith (1990) scheme to keep the cloud fraction in memory between model timesteps, and this alows the responses to sharp humidity gradients, for example in frontal systems, to be smoothed out. As in other convection-permitting UM simulations, we retain the older Smith (1990) scheme in our regional model as the pc2 scheme has not been extensively tested for resolutions higher than 12 km (Kendon et al., 2012).

## 3 Model tuning

We classify the parameters we changed in the model as aerosol tuning parameters (the hygroscopicity of organic carbon, the fraction of a raining grid box for aerosol removal and the dry deposition velocity) and cloud microphysical parameters (the autoconversion and accretion rates and critical relative humidity) and note them in Table 1.

First, we upgrade the refractive index of black carbon from the value set by the World Climate Research Programme (Deepak et al., 1983) to use the recommendation of Bond and Bergstrom (2006). This matches the latest GA7.1 configuration of the Unified Model (Walters et al., 2017a), and leads to increased aerosol optical depth in better agreement with observations (Stier et al., 2007).

The uncertainty analysis of Regayre et al. (2018) shows that the main driver of uncertainties in CCN concentrations in this region is the dry deposition velocity of the accumulation mode. The uncertainty in the parameter is deemed to be a factor 10 in either direction in their simulations. We find the global and regional model performance is slightly improved by increasing the dry deposition velocity by a factor of three, which gives a small (around 10%) reduction in background aerosol concentrations in the boundary layer.

While we found reasonably good agreement of CCN concentrations with observations after this modification to the dry deposition velocity, too little aerosol activated to reproduce MODIS cloud droplet number concentrations during the smoke





episode. Therefore, we tuned the width of the updraft velocity PDF and the hygroscopicity of the smoke aerosol to approximately reproduce the mean CDNC observed by MODIS. Initially, the updraft PDF width $\sigma_w$ was set to

$$\sigma_w = \max\left(0.1, \frac{2}{3}\sqrt{TKE}\right) \tag{1}$$

where $TKE$ is the boundary layer turbulent kinetic energy. We fixed it to $0.12\,\mathrm{m\,s^{-1}}$.

Aerosol hygroscopicity in the model is calculated on an average basis using Köhler theory assuming the aerosols are internally mixed. We used a kappa value (Petters and Kreidenweis, 2007) for organic carbon aerosol (OC) of 0.88, compared to 0.97 for sulphate, 0.0 for black carbon, and 0.99 for sea spray. This hygroscopicity for OC is unrealistically high if OC is considered in isolation, even for very aged biomass burning aerosol. Measured values of kappa in biomass burning aerosol tracers such as levoglucosan are generally around 0.1 or 0.2 (e.g. Mochida and Kawamura, 2004; Petters and Kreidenweis,
2007). This probably means our smoke aerosol particles are too small, or too low in concentration. However, the hygroscopicity of aerosol in the model could well be quite reasonable overall, since there are no sulphur emissions associated with biomass burning aerosol in the UKCA model, and in reality a moderate sulphate content due to co-emitted sulphur dioxide will make biomass burning aerosol more hygroscopic than our model would suggest. Global emissions of around $2\,\mathrm{Tg\,yr^{-1}}$ of sulphur dioxide due to biomass burning are reported by, for example, Schultz et al. (2008). Furthermore, our biomass burning aerosol
will become coated with sulphate (e.g. Khalizov et al., 2009) via condensation in the free troposphere or, once entrained into the boundary layer, also by aqueous processing in cloud droplets. These processes are included in our model, but they increase sulphur content throughout the bulk volume of the aerosols. In reality, processing and condensation coat the aerosol surface, and it is presumably the sulphate surface that controls activation, rather than the volume average of BC, OC and sulphate.

       The hygroscopicity needed to get reasonable closure between CCN and CDNC also depends on the aerosol size distribution
and on the updraft velocity. The lack of knowledge of the updraft velocity, and of how to scale it to get reasonable activation in a large-scale model with a parameterisation like that of Abdul-Razzak and Ghan (2000) makes it difficult to get the right answer for the right reasons. A dedicated study is beyond the scope of this paper, but some limitations of Abdul-Razzak and Ghan (2000) are discussed by Ghan et al. (2011), where it is compared with more sophisticated but much slower schemes, for example by Nenes and Seinfeld (2003).

We test two possible parameterisations for autoconversion and accretion in the warm rain scheme: that of Khairoutdinov and Kogan (2000) (the default in our simulations) and that of Kogan (2013). Insofar as it is represented in our model, this is a test of the 'second aerosol indirect effect' since it is via these parameterisations that rain rates are sensitive to aerosol. In the regional model, to ensure that aerosol particles are appropriately depleted by rain, we also change the value of the fraction of each grid box that is precipitating from 0.3 to 1.0. The factor 0.3 is appropriate in the 65-km grid boxes of the climate model
but not for the $4\,\mathrm{km}$ resolution model. This affects only the scavenging of aerosol, not the rain rate.



**Table 1.** Adjusted parameters in our simulations compared to the usual climate model. In the table, K & K refers to Khairoutdinov and Kogan (2000), and the change to the cloud scheme critical relative humidity and other parameters are defined in the text. GA6 is a standard configuration of the Unified Model described by Walters et al. (2017b). IMRI refers to the imaginary part of the refractive index of black carbon (BC); organic carbon is abbreviated by OC, and B & B refers to Bond and Bergstrom (2006).

| Parameter | GA6 | Global model | Regional model |
|---|---|---|---|
| Dry deposition velocity | 1 | Factor 3 higher | Factor 3 higher |
| Biomass burning emissions | GFED | FEER | FEER |
| Biomass burning diameter | 150 | 120 | 120 |
| Biomass burning emissions scaling | 2 | 1 | 1 |
| Rain fraction | 0.3 | 0.3 | 1 |
| BC IMRI | WCRP 1983 | B & B (2006) | B & B (2006) |
| Kappa value for OC | 0 | 0.88 | 0.88 |
| Updraft width (ms$^{-1}$) | TKE based/0.1 | 0.12 | 0.12 |
| Autoconversion and accretion rates | K & K | K & K | K & K or Kogan (2013) |

## 4 Satellite retrievals

We use cloud droplet number concentrations (CDNC) calculated from Moderate Resolution Imaging Spectroradiometer (MODIS) retrievals, and from the Spinning Enhanced Visible and Infrared Imager (SEVIRI) Cloud Physical Properties algorithm. The calculation of CDNC from cloud effective radius, optical depth, and cloud top temperature from MODIS Collection 6 (Platnick

et al., 2015) level 2 data is outlined in Supplementary Section 11. We use liquid water path (LWP) from MODIS and the Advanced Microwave Scanning Radiometer (AMSR-2) (Wentz et al., 2014) (level 2). MODIS retrievals with pixels identified as partly cloudy are removed using the 'clear sky restoral' logic (Platnick et al., 2015). We use aerosol optical depth for cloud-free scenes from MODIS, also collection 6 and level 2 (dark target) (Levy et al., 2015), and aerosol index from the Ozone Mapping Profiler Suite (OMPS) (Seftor and McPeters, 2017; Flynn et al., 2014). Precipitation fields are obtained from gauge-calibrated

Global Precipitation Measurement (GPM) Level 3 data (Huffman et al., 2014) at half-hourly time resolution and 0.1° spatial resolution. We also show feature masks from CALIOP Liu et al. (2005). It is not the purpose of this study to evaluate the accuracy of the satellite retrievals: we compare the most critical retrievals to get an idea of probable uncertainties, and assume the model is unlikely to be closer to reality than the retrievals. Nevertheless, it should be emphasised that the cloud retrievals are potentially unreliable in this area, because there is frequently aerosol overlying the clouds (e.g. Haywood et al., 2004) and

the cloud structure can be heterogenous at scales smaller than the 1-km MODIS grid.

   To help reduce effects due to cloud heterogeneity (e.g. Zhang et al., 2012), we use the 3.7 μm wavelength effective radius retrieval from MODIS to calculate CDNC and LWP. We also examine the effective radius, optical thickness and cloud top temperature retrievals that we use to calculate CDNC, and remove retrievals if the cloud top temperature is below 275 K, as such temperatures indicate that the satellite is seeing a high cloud rather than the stratocumulus deck of interest here.





The comparison of the CDNC and its component variables (cloud optical thickness, cloud top temperature and droplet effective radius) retrieved by MODIS and SEVIRI CPP (Deneke et al.; Roebeling et al., 2008b) is shown in Supplementary Figures S2, S3 and S4 on 2 and 7 August. The formulae used to calculate cloud droplet number concentration are presented in Supplementary Section 11, and summary statistics for the whole period are given in Supplementary Table S1. The SEVIRI

effective radius and optical depth are both low compared to MODIS, leading to a lower cloud droplet number concentration (where the differences compared to MODIS tend to cancel, but effective radius is more important than optical depth) and a substantially lower liquid water path (where the biases in the two retrievals add). This may be partly due to the different wavelength used for the retrievals ($1.6\,\mu$m for SEVIRI, $3.7\,\mu$m for MODIS). The algorithms used to calculate CDNC also play a role, since the SEVIRI CDNC is significantly lower than the MODIS CDNC even when the systematic differences

between the effective radii and optical depths are small, as on $8^{th}$ August (Table S1). The SEVIRI retrievals may be more strongly affected by broken cloud fields than MODIS retrievals as the SEVIRI pixel size is usually larger (a minimum of $3.5\,\mathrm{km}$ compared to a minimum of $1\,\mathrm{km}$ in MODIS, but the MODIS pixel size varies more across the domain due to the small swath width). However, biases in the two retrievals are still quite likely to be correlated because the basic technology is the same in the two cases. Since the spatial trends are broadly consistent and the quantitative agreement of the CDNC retrievals

is within a factor 2, and the spatial coverage is much larger than that of MODIS, we use the SEVIRI CDNC for a qualitative comparison with the global model in Section 5.

For liquid water path we can also compare MODIS and SEVIRI with AMSR-2, in pixels defined as cloudy (here, LWP $> 10\,\mathrm{g\,m^{-2}}$), on the coarser AMSR-2 grid. These retrievals are less likely to have correlated biases as MODIS uses visible and AMSR-2 microwave radiation. In particular, AMSR retrievals are less likely to be affected by overlying absorbing

aerosol (Bennartz, 2007; Bennartz and Harshvardhan, 2007). The normalised mean biases for each day are shown for both MODIS and SEVIRI with respect to AMSR-2 in Supplementary Table S2. We find that the MODIS retrievals are usually biased low compared to AMSR by around 10% (though occasionally up to 30%). This may be due to overlying aerosol, or it may be due to the inclusion of the rain water path in the AMSR retrieval. The SEVIRI retrievals are mostly biased low by 20-40%; as this is significantly larger than the value for MODIS, and larger than the uncertainty that would be estimated from

individual sensors, we do not use the SEVIRI retrieval for comparisons with the model.

## 5   Smoke transport in the global model

Smoke aerosol in our model can be traced by the mass mixing ratio of black carbon. This is shown above and below the stratocumulus inversion height (the highest altitude for which the relative humidity is above 60%, or $1300\,\mathrm{m}$ if there is no clear inversion) for the global model at midday on 2 and 7 August in Figure 2. The inversion height itself is also shown, but only

where a clear inversion is present. The figures show that the majority of the smoke is advected across the ocean above the boundary layer, and that much more of it mixes down into the boundary layer on 7 August than on 2 August. The right hand plots show how the height of the inversion increases as the sea surface temperature increases from east to west. Based on the



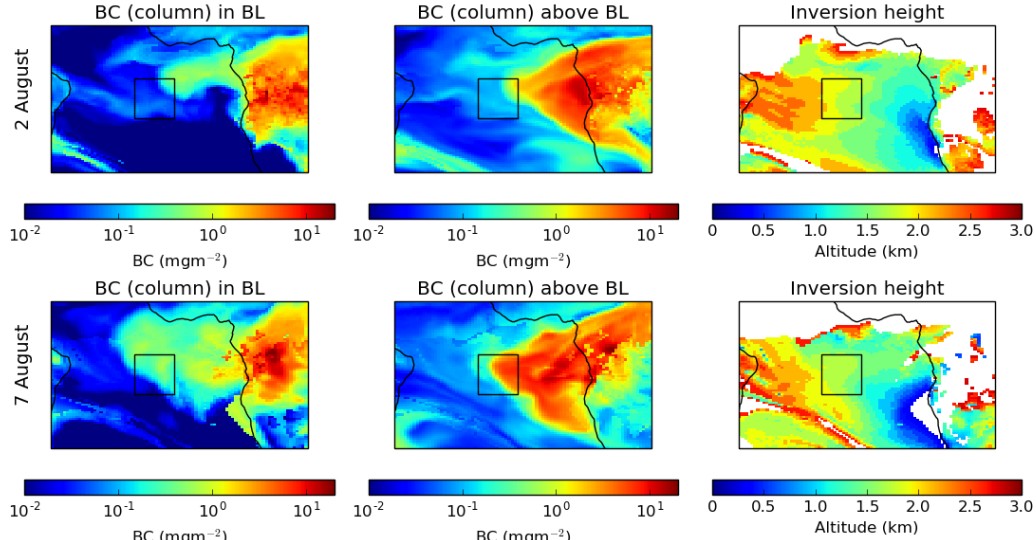

**Figure 2.** Black carbon (BC) column mass above and below the inversion (or above and below 1300 m altitude, if a clear inversion could not be diagnosed) in the global model on 2 August and 7 August, and the inversion height on these days. Where 1300 m altitude is used because there is no inversion, or where the inversion height is over 2800 m above the surface, the right hand plots are white. The marked square is the domain of the regional model.

timeseries we present later, we now define 1-5 August as the 'pristine period' and 6-10 August as the 'polluted period' for our simulation.

We used the horizontal and vertical wind fields from our global model in the LAGRANTO software package (Sprenger and Wernli, 2015) to produce back trajectories to track the smoke plume as it makes its way out to Ascension Island. Example
10-day trajectories ending at the corners and centre of our regional model domain at 1500 m altitude are shown at the top of Figure 3, for 2 and 7 August. This altitude is just below the top of the boundary layer in the domain of our regional model. On 7 August, the air masses cross the African continent, where they pick up smoke, while on 2 August most of them do not. The lower plots show the fraction of 8500 back-trajectories from points distributed evenly within the regional model domain that pass over the African continent, as a function of altitude. Around 20% of free-tropospheric trajectories that end on 2 August do
pass over the continent, but almost none in the boundary layer. The majority of trajectories at all altitudes that end on 7 August pass over Africa. During smoke episodes, it takes around four days for smoke aerosol to reach our regional domain (centred at $10°$S, $12°$W) from the African coast. The more detailed study of Adebiyi and Zuidema (2016) found that it took 7-9 days for smoke to reach $20°$W. Our analysis is consistent with this if we consider that $20°$W is around $3°$ further into the ocean than the right hand edge of our domain, and the smoke is emitted inland rather than at the coastline.
The large-scale behaviour of the global model is compared qualitatively with observations from SEVIRI and CALIOP to evaluate its ability to deliver the right aerosol concentrations to the regional model in Figure 4. More quantitative evaluation



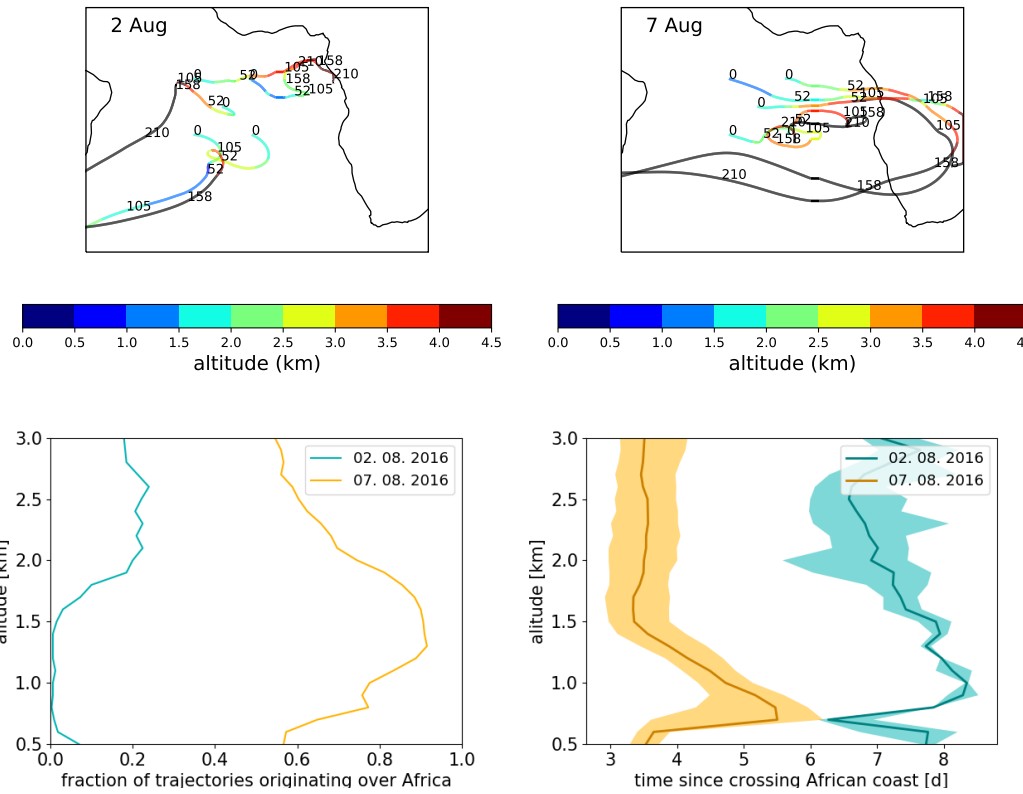

**Figure 3.** Trajectories (top) from the global model for 10 days prior to 2 August (left) and 7 August (right) calculated backwards from at the center and the four corners of the regional model domain at 1500 m altitude. The numbers along each path correspond to the number of hours backwards along the air mass trajectory. Below, the fraction of back-trajectories on 2 August (blue) and 7 August (orange) that pass over the African continent in the ten days before they reach the regional model domain are shown. The right hand plot shows the time taken for the trajectories that do cross the continent to reach the region on the two dates. The shaded bands show the interquartile range. At each of 25 different altitudes, 330 sample trajectories were calculated, spaced evenly across the model domain. This is a large sample, but the interquartile range for the case of 2 August is limited by the sample size below 1.5 km altitude because so few trajectories come from Africa.





of the microphysics in the global model using MODIS retrievals in the region of interest is described later, in comparison with the regional model. Figure 4 shows, at the top, the CDNC observed by SEVIRI and that in the global model, on 2 August (left) and 7 August (right). The global model values are calculated for the grid box at cloud top: the highest grid box in the model below $3\,\mathrm{km}$ altitude with a liquid water mixing ratio above $1 \times 10^{-5}\,\mathrm{kg/kg}$. The model appears to overestimate CDNC

compared to SEVIRI on 2 August in the mid-Atlantic and near Gabon, and underestimate them close to the Namibian and southern Angolan coasts. However, the large-scale trends (lower CDNC in the south-east corner, higher in the north-east, and variability in between depending on the weather systems) are well captured. On 7 August, elevated CDNC are seen by SEVIRI in a large region centred on the western edge of the stratocumulus deck. This is replicated in the model, although the cloud fraction in the model is higher than in the observations north-east of the centre of the sub-figure. Also, the area of high CDNC

extends further west in the model than in the observations. This suggests the model either simulates faster advection of the plume across the ocean than reality, or more mixing of the plume into the clouds.

The CALIOP feature masks are used to evaluate the vertical distribution of clouds and aerosols. The masks show land in yellow, aerosol in light blue and cloud in dark blue. Only the cloud-aerosol mask is shown. Supplementary Figure S1 shows that, over this region of the Atlantic Ocean on one of our simulation days, the feature mask will detect an aerosol layer if the

AOD measured by CALIOP is above around 0.03. This is comparable to the mean background AOD in the simulations (see Figure 8, discussed later). In Figure 4, the black carbon mass mixing ratio in the model is shown next to the feature masks for the same cross-section. The simulated cloud top height is shown on these plots as a red line, and the cloud base is in orange.

The comparison in Figure 4 shows that mixing of the aerosol down into the boundary layer is clearly reproduced by the model, and the observed and modelled cloud height are in good agreement, to within one vertical grid level of the global

model, or about $200\,\mathrm{m}$. The height of the peak of the aerosol layer in the model is reasonable but appears rather too low compared to the feature mask, usually by $1 - 1.5\,\mathrm{km}$. The reason for this is not currently understood in detail: it may be due to the overestimation of the height of the aerosol layer in CALIOP noted by Painemal et al. (2014) and Rajapakshe et al. (2017), or to too much subsidence, a common feature among general circulation models (Das et al., 2017), or some other reason, perhaps atmospheric circulation patterns (Adebiyi and Zuidema, 2016). It is noted by Das et al. (2017) and Adebiyi

and Zuidema (2016) that ERA-interim, which we use to nudge our model, performs better than other reanalyses.

## 6   Regional model evaluation and comparison of regional and global models

### 6.1   Meteorology and dynamics

The basic dynamics of the regional and global models are evaluated by comparison to Vaisala radiosondes launched from Ascension Island as part of the LASIC campaign (D. Holdridge, J. Kyrouac and R. Coulter, 2016b). The model grid box

closest to the sonde in the horizontal is chosen, then as the sonde takes a large number of readings at each vertical level in the model, vertical interpolation is performed to obtain the temperature in the model at the location of every $20^{th}$ sonde reading. The colour plots in Figure 5 show that when the whole timeseries of soundings is considered, the variability in temperature and RH in the observations at the level of the clouds is represented better in the regional model. This is simply because the smaller





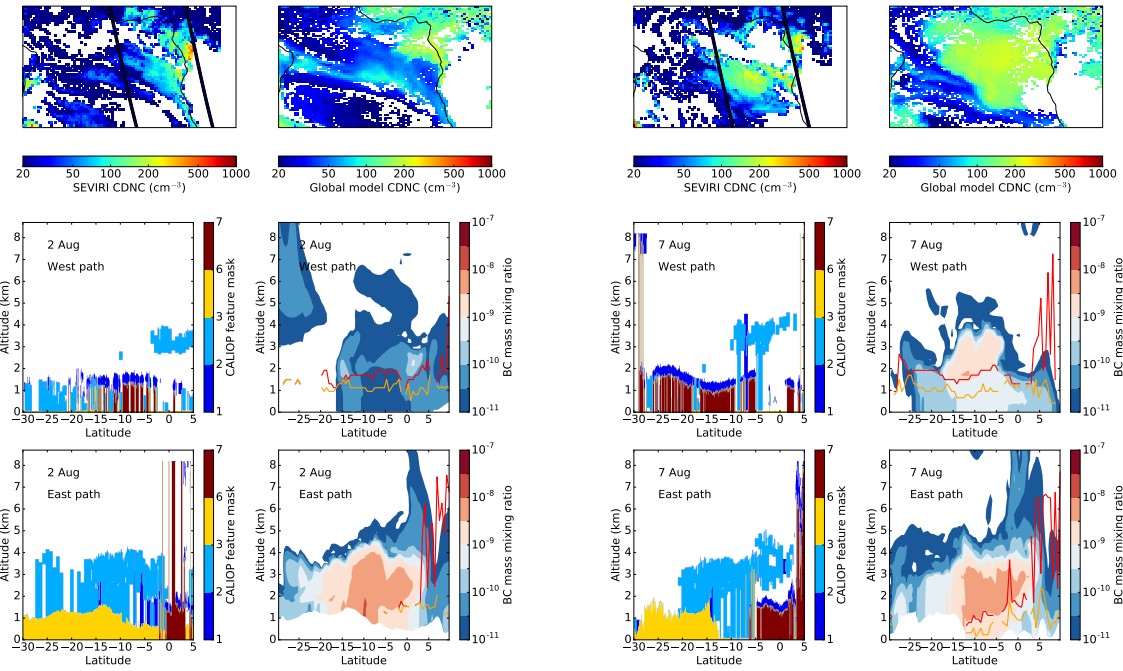

**Figure 4.** Evaluation of aerosol and cloud in the global model. Cloud droplet number concentrations from the SEVIRI satellite instrument and from the model are shown on the top row. In both the model and the observations, grid boxes with liquid water path less than $10 \, \mathrm{g \, m^{-2}}$ are assumed to be cloud-free and are excluded, after degrading the resolution of the satellite to be comparable to the model. The left column shows SEVIRI CDNC, and, below this, the CALIOP feature mask for the marked easterly and westerly overpasses on 2nd August (as for the trajectory plots). The second column shows the model CDNC and, below this, the simulated black carbon mass mixing ratio, used here as a proxy for smoke aerosol. The red line denotes the altitude of the cloud top and the orange line that of cloud base. The right two columns show the same, but for 7 August. In the feature mask, 1-2 (dark blue) indicates cloud, 2-3 (light blue) aerosol, 3-6 (yellow) land above sea level, and 6-7 (dark red) indicates that the signal is fully attenuated.





grid boxes resolve more fluctuations in the cloud field. However, the skill of the regional and global models is very similar. The bottom subplots of Figure 5 show ten example profiles for 1, 3, 5, 7, and 9 August 2016.

The temperature profiles show the strong inversion characteristic of this region. The inversion strength varies from 5 to 9 K. The height of the inversion is generally captured by the model to within about 300 m, though it is significantly overpredicted on 8 and 9 August, especially at night, and smeared out by the model resolution as expected. On 10 August (not shown) good agreement with the observations is recovered. The temperature gradient within the inversion averages $0.027 \, \text{K m}^{-1}$ in the radiosonde data, $0.013 \, \text{K m}^{-1}$ in a representative sample of the regional model output, and around $0.010 \, \text{K m}^{-1}$ in the global model (see Supplementary Figure S6). Below the inversion, the temperature profile in the model is very close (usually within 3 K) to the observations. However, above the inversion, some of the structure in the observations, in particular many of the small inversions at higher altitude, are not always captured by the model. These inversions may separate layers of air that have followed different trajectories, and due to the limited number of observations the ERA-interim reanalysis can draw on, and its relatively low vertical resolution, these different trajectories may not always be simulated accurately.

The relative humidity in the model matches the observations quite well, within 20% almost everywhere in the atmospheric column, at the beginning of the simulations when air masses are mostly of marine origin. However, between 2 and 5 August, the humidity above the boundary layer is consistently higher in the observations than in the model, sometimes by as much as 40%. As suggested by Adebiyi et al. (2015), this is associated with elevated aerosol concentrations at this altitude (see Section 6.2). At the end of the simulation period, the height of the boundary layer differs more between the model and the radiosondes, and therefore the discrepancy in relative humidity is larger.

### 6.2 Aerosols: cloud condensation nuclei and optical depth

The CCN concentrations in the global and regional model are compared to observations from the ARM site at Ascension Island (D. Holdridge, J. Kyrouac and R. Coulter, 2016a) on the left side of Figure 6. A reasonable level of agreement between the model and observations of both CCN and CDNC would provide some confidence in our treatment of activation. While the mean concentration in the pristine part of the simulation matches the observations well, there is some structure that is not fully captured by the model. The occasional extremely low CCN concentrations recorded at Ascension are likely due to wet scavenging of aerosol. Neither the regional model nor the global model can capture this fully, though the regional model does have some more modest fluctuations that are also probably the result of wet scavenging. In our setup, the *cloud microphysics* code simulates liquid water and precipitation, and receives as an input the cloud droplet number concentration from the *aerosol microphysics* code. In the aerosol microphysics code, the fraction of aerosol that remains after a time step of length $\Delta t$ in which wet scavenging is occuring is equal to $e^{-\Delta t A/L}$ where, for warm clouds, $A$ is the sum of the autoconversion and accretion rates and $L$ is the liquid water content, all fed in as inputs from the cloud microphysics code. This is something of a simplification: it performs adequately on average but not when high spatial and time resolution are needed. It could be improved in future by simulating the removal of aerosol entirely in the cloud microphysics scheme: aerosol mass and number concentrations would be given to the cloud microphysics code, activated to (prognostic) cloud droplets, depleted by wet scavenging or transported via sedimentation of rain droplets, and then passed back to the aerosol microphysics.



**Figure 5.** Temperature and relative humidity at Ascension Island compared with ARM radiosondes (D. Holdridge, J. Kyrouac and R. Coulter, 2016b). The top six plots show the observed (left), and the regional (centre) and global (right) model time-series of temperature (above) and relative humidity (below). The next five profile plots are representative of night-time, and the bottom five of the afternoon, on 1,3,5,7,9 August. The date and time of the sonde release is shown bottom-left.




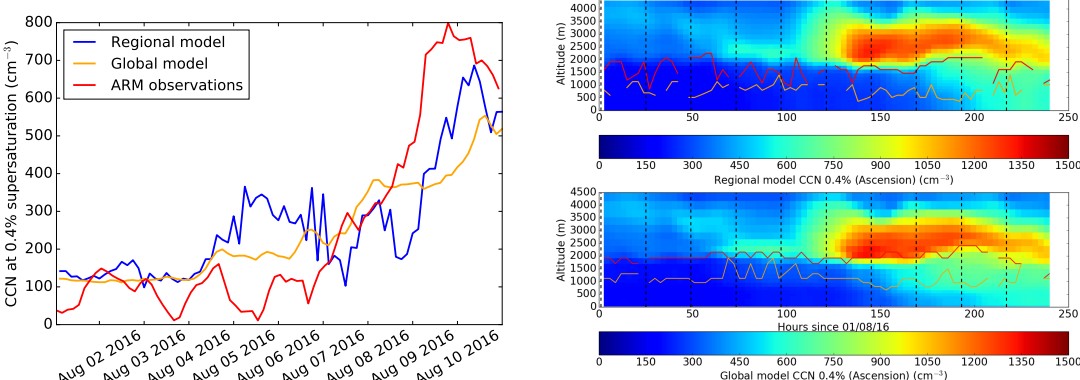

**Figure 6.** Left, CCN at 0.4% at Ascension at the surface measured with the ARM CCN counter (D. Holdridge, J. Kyrouac and R. Coulter, 2016a) and compared to our simulations, and right, the vertical profiles of CCN in the global and regional models, averaged over the domain. The cloud top and cloud base heights at Ascension Island itself are marked in red and orange respectively. The black dotted lines indicate midnight, local time.

Across the first four days, the mean and standard deviation of the CCN concentration at 0.4% is $90 \pm 42\,\mathrm{cm^{-3}}$ in the observations, $148 \pm 40\,\mathrm{cm^{-3}}$ in the regional model and $132 \pm 28\,\mathrm{cm^{-3}}$ in the global model. The modelled and observed values are samples of three-hourly means. The onset of smoke mixing down into the boundary layer is approximately correct compared to observations, but around half a day later on 7 August in the regional model than the global model. The peak CCN concentration

is underestimated, but correct to within around 50%.

The right-hand side of Figure 6 shows the vertical profile of CCN at 0.4% supersaturation in the regional and global models, averaged over the domain. The particle concentrations are in good agreement between models. Between cloud top and $3500\,\mathrm{m}$ altitude, which coversthe smoke plume (Figure 6), the mean and standard deviation of the CCN 0.4% concentration in the last three days of the period is $1032 \pm 353\,\mathrm{cm^{-3}}$ in the global model and $992 \pm 352\,\mathrm{cm^{-3}}$ the regional model irrespective of the

cloud. Likewise, between the surface and the top of the clouds, the regional model mean is $506 \pm 197\,\mathrm{cm^{-3}}$ while the global model mean is $584 \pm 169\,\mathrm{cm^{-3}}$. The small differences in concentration are likely due to differences in the boundary layer scheme and vertical resolution between models, and to advection. The wind fields in the regional model are only nudged to ERA-interim at the boundaries of the domain, so they start to deviate from the reanalysis towards the end of our simulation period.

The differences in the mean and standard deviation of the CCN concentration between the regional and global models are small. The model resolution clearly has some effect on the scavenging of aerosol by precipitation. This is illustrated by the snapshots of CCN at 0.4%, shown for the whole domain but corresponding to the same altitude ranges (surface to domain mean cloud top, and mean cloud top to $3500\,\mathrm{m}$) in Figure 7. The left hand plots correspond to the time of the AQUA overpass on 7 August and the right-hand plots to 00:01 UTC on 10 August. On 9 August, the spatial pattern of CCN concentration



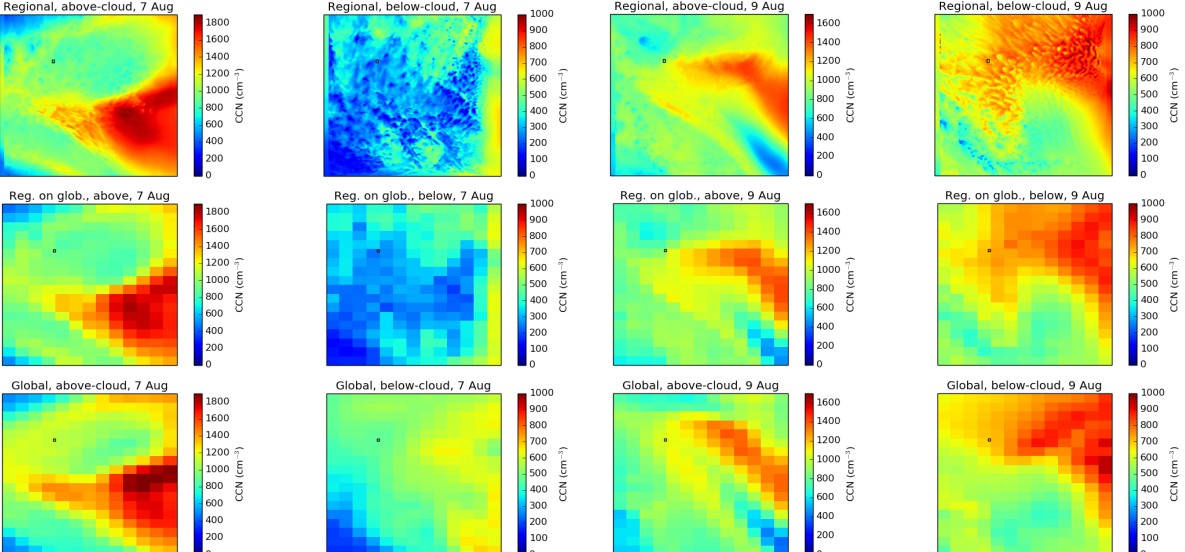

**Figure 7.** Snapshots of CCN at 0.4% supersaturation across the regional model domain, in the regional and global models. The left six plots show the time of the AQUA overpass on 7 August, also shown for the regional model in Figure 10. The right six plots show 00:01 UTC on 9 August, when the aerosol plume covers different parts of the domain. Within each group, the top plots are for the regional model and the bottom for the global model, left between the domain mean cloud top height and 3500 m altitude, and right, below the domain mean cloud top height. Note that clouds extend slightly above the domain mean cloud top height in the western half of the domain. The centre plots are for the regional model, regridded to the same resolution as the global model. The small black square represents Ascension Island.

again suggests that the wind fields in the regional model are deviating from those in the nudged global model. The effect of precipitation is to deplete large areas of the global model quite uniformly, but to deplete small pockets in the regional model. The small differences in the mean concentrations of CCN between models may be due to deposition, transport across the domain, or different boundary layer mixing. Both the boundary layer scheme and the sub-grid cloud scheme differ between models (contributing to different cloud fractions, described later). The global model can also transport aerosol in the convection scheme. To clarify exactly what is responsible for the different mixing, future studies will investigate aerosol scavenging in identical regional setups with different resolutions, to see whether the higher horizontal resolution makes a difference to the mean aerosol concentrations independently of the model configuration.

Figure 8 shows a comparison of the aerosol optical depth (AOD) in the model with MODIS 10-km retrievals, averaged across the regional domain. The mean aerosol optical depth in our regional model over the last five days of the simulation is 0.35, and in the global model it is 0.45, while in the observations it is 0.40. The aerosol optical depth is a factor 3.2 higher in the observations than in the regional model in the first half of the simulation period; this discrepancy is already expected from the relative humidity profiles on the assumption that elevated aerosol and elevated moisture occur together. The smoke plume seems to arrive slightly earlier in reality than in the simulations, too, perhaps because the model places the smoke slightly




too low in altitude compared to the CALIOP observations. The profiles in Supplementary Figure S7 show the wind is usually stronger at 3000 m, the level of the aerosol layers seen by CALIOP, than at 2000 m, where much of the aerosol appears in the model.

On the right of Figure 8, we show the aerosol index from OMPS, a complementary satellite retrieval of aerosols. This is averaged over a $7° \times 7°$ box centred on Ascension Island. Aerosol index (AI) has been combined with MODIS retrievals to derive AOD above clouds (e.g. Torres et al., 2012). We do not repeat this here, but the raw UV AI still yields complementary information to the AOD (Wilcox, 2010): first, clouds are screened out of the AOD retrieval but not the AI retrieval, and second, the AI is better correlated with aerosol number concentrations (Nakajima et al., 2001; Costantino and Bréon, 2013). Our timeseries (Figure 8) shows the average of all valid aerosol index retrievals in our domain on a given day; the main criteria for excluding data is if the sun-glint angle is less than $20°$, the solar zenith angle is above $70°$, or the aerosol index is below 0.5 (Seftor and McPeters, 2017). This last threshold will preferentially select absorbing aerosol, as this tends to have positive aerosol index (Hammer et al., 2016). Qualitatively, it is in generally good agreement with the aerosol optical depth.

Both the timeseries of aerosol index and aerosol optical depth reinforce our conclusion that the smoke aerosol arrives earlier over Ascension Island in reality than in our simulations, but at high altitude so it is not immediately entrained into the boundary layer. The aerosol then mixes down into the boundary layer a day later in the observations than in the simulation, so that the CCN concentrations in the observations at Ascension Island rise later than they do in the model. The peak relative humidity above the boundary layer is superposed on the right hand plot in Figure 8, and it is clear that high aerosol is strongly correlated to high humidity (Adebiyi et al., 2015). In Supplementary Figure S5, we show that this is also true in the model, at least well above the boundary layer: high humidity, coincident with aerosol, can be seen above the inversion in Figure 5 on 5, 7, and 9 August.

### 6.3 Cloud droplet number concentration and liquid water path

Figure 9 shows the time series of domain-median cloud top droplet number concentration (CDNC) and in-cloud liquid water path (LWP) in the regional and global models, compared to means from MODIS and, for LWP, AMSR. In the case of MODIS, the satellite data are regridded to the coarser regional model grid, while in the case of AMSR the regional model data are regridded to the coarser AMSR grid. The global model data are not regridded. In both models and observations, a pristine period and a polluted period can be identified, but the CDNC rises in the regional model before it rises in the observations, again indicating too-early mixing into the boundary layer. However, from 1-4 August and 7-10 August, good agreement (to within $50\,\mathrm{cm}^{-3}$) between modelled and observed CDNC is apparent. The normalised mean bias of the timeseries of regional mean (not median) LWP is given by

$$100\% \times \frac{\sum (LWP_m - LWP_o)}{\sum LWP_o} \qquad (2)$$

where the sum is over the data in the time-series and the subscript $o$ refers to observed values and $m$ to simulated values. It is 22% in the regional model and 66% in the global model. The overestimate of liquid water path in the global model is quite uniform over the simulation period while in the regional model it is particular to the 6-8 August period. In the regional model,





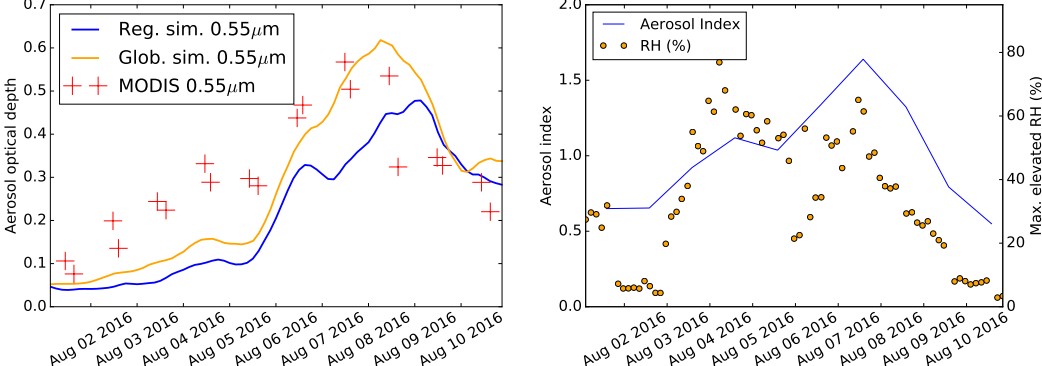

**Figure 8.** Left: aerosol optical depth, averaged over the regional domain. We did not attempt to correct sampling biases: retrievals in the MODIS aerosol optical depth domain average are not given greater weight when they are further away from other retrievals, so cloudy areas where there are few AOD retrievals do not contribute as much to the domain average as areas where the skies are clear and there are many retrievals. As large areas of the regional domain have no retrievals because they are cloudy, at the edge of swaths, or in the glint regions, this bias is probably significant, but it is likely to explain some of the scatter, rather than the trend, shown in the figure. Right: OMPS aerosol index in a $7 \times 7°$ latitude/longitude box centred on Ascension Island and maximum relative humidity above the boundary layer, measured by the ARM site sondes (D. Holdridge, J. Kyrouac and R. Coulter, 2016b).

it may be related to the overestimate of the boundary layer height at this time (Figure 5). The Pearson correlation coefficient between the domain mean simulated and observed timeseries is 0.60 for the regional model and 0.56 for the global model.

The characteristics of the cloud field are shown in Figure 10. On the left, we present histograms of CDNC and LWP in the regional domain, for the regional model, the global model (normalised to the number of regional model grid boxes) and

5 MODIS AQUA observations. Pixels with liquid water path below $10 \, \mathrm{g \, m^{-2}}$ are screened out. On the right, the cloud-top CDNC and LWP fields are compared between MODIS and the regional model. There is a small area of cirrus on 2 August visible in the MODIS retrievals in the bottom-left of the regional domain (screened out of the CDNC but not the LWP). On 7 August, the MODIS cloud top temperature sometimes corresponds to heights well above the boundary layer (around $240 \, \mathrm{K}$ - see Supplementary Figure S2) due to high cloud or possibly the aerosol layer, and so some of the CDNC retrievals in the lower

half of the regional domain had to be screened out.

During the pristine period (1-5 August), the features of the CDNC distribution, including structures that appear to be cellular convection, are reasonably well captured by the model. The slightly more polluted region to the south of Ascension is mirrored in the model data, but not the higher CDNC in the broken clouds to the east of the island. In the polluted period on 7 August, as well as a 50% overestimate of CDNC by the model compared to MODIS due to too early mixing, the CDNC (Figure 10)

also seem less homogenous in the observations than in the model. This may be due to rain events outside the model domain that depleted the aerosol on its way out to Ascension Island.



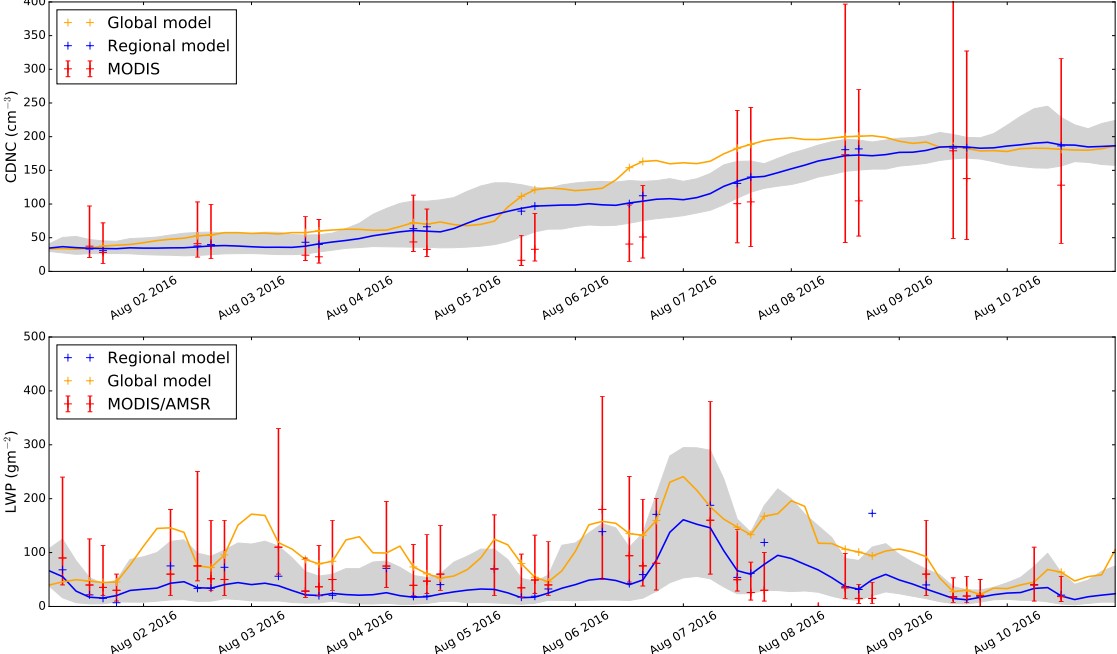

**Figure 9.** Time series of cloud droplet number concentration at cloud top and in-cloud liquid water path in the model (a median within approximately $800\,\mathrm{km}$ of the centre of the domain) compared to corresponding AMSR and MODIS observations across the same area at matched times. The MODIS data were regridded to the resolution of the regional model. Cloudy grid boxes are defined as those with LWP above $10\,\mathrm{g\,m^{-2}}$. The interquartile ranges in the satellite data are shown as error bars, and the range in the regional model as a grey band. Where the blue crosses deviate from the overall domain median indicated by the blue lines, the regional model data were regridded to the AMSR resolution and averaged only inside the AMSR swaths, or calculated only inside the MODIS swaths. Thus the blue and red crosses are directly comparable. No correction for swaths was made in the lower resolution global model data, hence the orange crosses lie on the orange lines and the blue and red crosses are only approximately comparable with the orange crosses. The effects of smoke aerosol on these quantities are presented and discussed later, in Figure 15.

The histograms in Figure 10 show that overall the distribution of LWP is well represented in the regional model, and the 2D plot shows that most clouds extend over similar spatial scales in the model and observations. We also calculated Moran's I autocorrelation metric (Moran, 1950) for the model and the MODIS AQUA observations (on the model grid) of liquid water path (where the regional domain is entirely within one satellite swath). This metric would be zero when liquid water is randomly distributed through the domain, and one when it is uniformly distributed across half of the domain with none elsewhere. On 2 August, it is 0.82 in the MODIS observations and 0.86 in the model, while on 7 August it is 0.87 in the satellite data and 0.90 in the model. In general the model tends to have fewer, larger, clouds (or cloud decks) than the satellite observes. We clustered adjacent pixels with liquid water paths above $50\,\mathrm{g\,m^{-2}}$, also on the model grid in both cases. The median size of these high-water-content clusters in the MODIS observations is usually 3 or 4 model grid boxes (12 or 16 square kilometres),





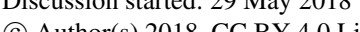

**Figure 10.** CDNC and LWP from MODIS AQUA and the model on 2 August (top) and 7 August (below). Wind vectors at 1260 m altitude are shown on the CDNC plot. On 7 August, the lower half of the MODIS CDNC retrieval is compromised by high cloud or aerosol, and the white areas in the CDNC plot that are not present in the LWP plot have been screened out because the cloud top temperature indicates cloud top heights are above the boundary layer.





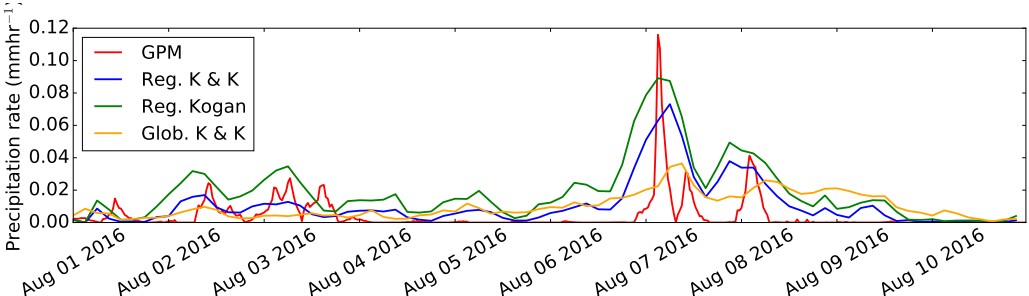

**Figure 11.** Domain mean precipitation rate in the regional and global models, in mm/hr, compared with the GPM dataset (Huffman et al., 2014). The effect of the biomass burning aerosol on rain is shown later, in Figure 14.

while in the model, the median number of grid boxes in these clusters at the time of the MODIS overpasses averages 15.6 over the simulation period, or nearly $70 \, \text{km}^2$. Despite these differences in the distribution of cloudiness, the figures show that there is some hope of encapsulating the most important area of the stratocumulus-to-cumulus region, where the clouds break up, inside the model domain. More detailed studies of this transition will follow in a subsequent paper.

The cloud fraction in the model is defined by the sub-grid cloud scheme, which differs between regional and global models. A qualitative comparison to the cloud fraction from the cloud scheme to the MODIS 1-km cloud mask (when this is regridded onto the model grid) would be possible, but the criterion for determining the cloud fraction (broadly, whether the retrieved cloud optical depth in a pixel is greater or less than 0.4), is not directly comparable to the model cloud fraction, and may also be sensitive to above-cloud aerosol (Platnick et al., 2015). We adopt the simpler approach of comparing the outgoing shortwave

radiation to observations from the CERES instrument instead, in Section 9. The regional and global model cloud fractions can be compared more directly by coarse-graining the regional model to the resolution of the global model and assuming the overall cloud fraction is the maximum cloud fraction in any given vertical column (as the vertical resolution is not the same in the two setups). The average cloud fraction in the global model is 0.60 during the first, clean, five days of the simulation period, while the regional model average cloud fraction is 0.54. During the second, polluted half of the simulation period, the mean

cloud fractions in the two models are 0.62 in the global model and 0.63 in the regional model. The spatial distributions of cloud fraction are shown in Supplementary Figure S8 and the timeseries in Supplementary Figure S12. When the cloud fractions are all averaged to the global model grid, the global model fraction is more likely to be at the extrema of 0 or 1 than the regional model. Histograms demonstrating this difference are shown in Supplementary Figure S8.

### 6.4 Precipitation

Figure 11 shows that the rain rates in the model are similar to retrieved rain rates from the gauge-calibrated GPM dataset. Good agreement of the model and observations is important as rain is associated with the transition from stratocumulus to cumulus clouds (Paluch and Lenschow, 1991; Yamaguchi et al., 2017). The model captures the patterns in the observational timeseries well, but overestimates the total precipitation amount and seems to precipitate for longer than is observed.



We tested two parameterisations of autoconversion and accretion rates, that of Kogan (2013), and that of Khairoutdinov and Kogan (2000), which is the default in our model. The two schemes yield significantly different rain rates (Figure 11). The Kogan (2013) parameterisation is more sensitive to aerosol: the autoconversion rate is proportional to

$$q^{4.22} N_d^{-3.01} \tag{3}$$

where $q$ is liquid water content and $N_d$ is cloud droplet number concentration, compared to

$$q^{2.47} N_d^{-1.79} \tag{4}$$

in Khairoutdinov and Kogan (2000). The normalised mean bias of the timeseries of the simulated rain rates with respect to the GPM dataset, calculated using Eq. 2, is +121% for Khairoutdinov and Kogan (2000) compared to +288% for Kogan (2013). In the case of the Kogan (2013) scheme, the increased rain rate acts to eliminate, on average, the bias of $+22\%$ in the

liquid water path observed with Khairoutdinov and Kogan (2000), which is reduced to -2%. Hill et al. (2015) also found that the Kogan (2013) parameterisation produced more rain than the Khairoutdinov and Kogan (2000) parameterisation. It is likely that the Khairoutdinov and Kogan (2000) accretion rates are more appropriate for this study, but the overall rain rates do not suggest which dependence on $N_d$ is the better. Within the domain (not shown), the distribution of rain is very similar in the two cases: the histogram of non-zero rainfall rates across the domain at any given time is simply scaled up with the Kogan (2013)

compared to the Khairoutdinov and Kogan (2000) parameterisation. The correlation of the observed domain-mean timeseries with the modelled timeseries is 0.56 for Kogan (2013), 0.56 for Khairoutdinov and Kogan (2000) and 0.25 for the global model (where the normalised mean bias is +105%).

Comparing to another regional modelling study, we simulate similar precipitation rates to Lu et al. (2018) at midday UTC: their model produces an average of $0.015 \, \mathrm{mm \, hr^{-1}}$ of rain at this time, while ours produces $0.014 \, \mathrm{mm \, hr^{-1}}$ with the Khairout-

dinov and Kogan (2000) scheme but $0.022 \, \mathrm{mm \, hr^{-1}}$ with the Kogan (2013) scheme. The spatial domain and averaging periods are not the same so this good agreement could be coincidental: we expect that in our spatial domain Lu et al. (2018) would predict a higher rain rate than their domain average, because their domain includes much more of the mainly non-precipitating stratocumulus deck than ours.

At the ARM site on Ascension Island, a total of $6.7 \, \mathrm{mm}$ of rain fell during the first five days of this period, and $3.8 \, \mathrm{mm}$

during the second five days. This is substantially higher than both the GPM and model domain mean results, but within $20 \, \mathrm{km}$ of Ascension Island, GPM is in very good agreement with the rain gauge: the dataset indicates $7 \, \mathrm{mm}$ of rain in the first five days and $3.9 \, \mathrm{mm}$ in the second. The regional model with the Khairoutdinov and Kogan (2000) parameterisation has $1.5 \, \mathrm{mm}$ in the first five days and $3.0 \, \mathrm{mm}$ in the second, and the Kogan (2013) parameterisation has $3.2 \, \mathrm{mm}$ in the first five days and $4.0 \, \mathrm{mm}$ in the second. One possible explanation for the higher rainfall observed at the island compared to the domain mean

is that Ascension Island is situated to the north-west of the centre of our regional domain, so the stratocumulus-to-cumulus transition may lead to more rain there than on the domain average. The local orography may also lead to more rain, as the site of the gauge is about $300 \, \mathrm{m}$ above sea level; this will not be represented in the model.





## 7   Effects of biomass burning aerosol on heating and temperature

In our simulations, we find biomass burning aerosol close to or just above the inversion causes short-wave heating and long-wave cooling close to the temperature inversion in the Ascension Island region, and the aerosol may thus affect the dynamics of the boundary layer. In the following section we compare three simulations: our default, a simulation with aerosol absorption

due to black carbon switched off so only the direct microphysical effects of the aerosol on the clouds are simulated, and a simulation without biomass burning aerosol. Aerosol absorption is switched off by setting the imaginary part of the refractive index of black carbon to zero (organic carbon does not absorb in the Unified Model). The sea surface temperatures in our simulations are fixed. Each regional simulation is driven by a different, paired global simulation: for example, the global simulation without absorption drives the regional simulation without absorption. The cloud response to the aerosol in the

regional model is therefore influenced by the response of the global model to the same aerosol on its way across the ocean towards the regional domain. We calculate heating rates from the divergence of the upwelling and downwelling radiative fluxes, and show them for our default regional model in Figure 12 and Supplementary Figure S9.

Substantial short-wave heating can be seen above the cloud layer during daytime towards the end of the simulation. This is mostly, but not entirely, offset by long-wave cooling, which has very little diurnal cycle, as shown in the middle plots of

Supplementary Figure S9. The result is a net cooling above the clouds during the night and early morning, and then a brief warming period at midday and early afternoon above the clouds. The daytime warming below the cloud layer, and night-time cooling at the top of the cloud layer (Supplementary Figure S9) is typical of this kind of cloud.

The heating rates due to the biomass burning aerosol are isolated by subtracting the heating rates in the simulations without biomass burning from those in simulations with biomass burning, in the central subfigures of Figure 12. The figure shows that

biomass burning aerosol causes strong heating at and above cloud top, and night-time cooling further aloft. A key effect of the biomass burning aerosol is to reduce the height of the cloud top (the red solid line in Figure 12 is below the red dashed line). This has been noted in satellite observations from CALIPSO by Wilcox (2010), who found the altitude of the peak of the cloud feature distribution is lower by up to $200 \, \mathrm{m}$ in the presence of smoke. The reduction in the cloud top height causes increased long-wave cooling at the new cloud top height, and reduced long-wave cooling at the old cloud top height. As

observed by Wilcox (2010), the cloud also gets thicker, as the cloud top rises while the cloud base height decreases. However, the WRF model of Lu et al. (2018) produces an increase in cloud top height in the presence of biomass burning aerosol, and the authors attribute the observed reductions to co-variability of aerosols and meteorology.

The short-wave heating effect of the aerosol layer, averaged between $2200 \, \mathrm{m}$ and $3500 \, \mathrm{m}$ over the last five days of the simulation, is $0.08 \, \mathrm{K \, hr^{-1}}$, for a mean AOD of 0.35. This range of altitudes is chosen to avoid the area just above the cloud top

where the short-wave effect of changing cloud top height also contributes to the overall short-wave effect. As this corresponds to $1.9 \, \mathrm{K \, day^{-1}}$, the rate is very similar to the $2 \, \mathrm{K \, day^{-1}}$ found in aerosol layers with AOD 0.4 in the Namibian stratocumulus deck by Wilcox (2010).

When we average instead over all altitudes up to $4700 \, \mathrm{m}$, the net heating rate (short-wave and long-wave) due to biomass burning aerosol is $+0.33 \, \mathrm{K \, day^{-1}}$, and this leads to an overall temperature increase in this part of the atmosphere of $+0.41 \, \mathrm{K}$





**Figure 12.** Regional average shortwave heating rate (top) and changes to heating rate and temperatures when biomass burning aerosol is introduced to the regional model (centre) and when aerosol absorption of radiation is switched on (bottom). The domain mean cloud top and cloud base heights in the simulations used are marked on the plots. On each day, midnight local time is indicated by a black dotted line.





over the last five days. A complex distribution of local temperature changes is shown on the right of Figure 12. The large ($\sim 6\,\mathrm{K}$) localised changes at or just above the cloud level broadly match the heating rate differences. There is a small warming (at most $1.5\,\mathrm{K}$) of the boundary layer, and overall cooling well above the inversion. However, the main effect of the aerosol is the heating above cloud top, which increases the stability of the system.

Comparing our default simulation with our simulation in which aerosol absorption is switched off, we obtain the bottom subfigures of Fig. 12. The indirect effects of the aerosol on the clouds are the same in both simulations. We infer that the heating rate due to aerosol absorption is $0.34\,\mathrm{K\,day^{-1}}$ and the net temperature change over the last five days of the simulation due to absorbing aerosol is $+0.52\,\mathrm{K}$ in the first $4700\,\mathrm{m}$ of altitude. The heating effect of the aerosol absorption is clearly visible just above the clouds. The effect of biomass burning aerosol and the effect of switching on aerosol absorption are very similar:

the central and bottom subfigures of Fig. 12 are almost identical. This confirms that the absorbing effects of the smoke aerosol are responsible for the heating rates, and that the smoke aerosol is the only absorbing aerosol present in the simulation.

Our simulated heating rates are unlikely to match the heating rates that actually occurred during this period, because the aerosol in our simulations is somewhat lower in altitude. This may mean the effect of the aerosol on the inversion for a given optical depth is exaggerated.

The vertical profiles of potential temperature shown in Figure 13 also show that the inversion is consistently lowered by the biomass burning aerosol on 7 and 9 August. The figure also shows large increases in liquid water content and a thickening of the cloud layer when biomass burning aerosol is present, and we discuss these in the following sections. Concerning the dynamical effects, Johnson et al. (2004) found (experiment 2-FT) that the absorbing effect of the aerosol above the clouds also tends to reduce the inversion height: the depth of the boundary layer changed by $50\,\mathrm{m}$ in their LES models. Locally, the picture

is more complicated, and the domain mean may hide significant north-south or east-west trends in inversion strength that may shed light on the stratocumulus-to-cumulus transition. For example Bretherton and Wyant (1997) found "that decoupling is mainly driven by an increasing ratio of the surface latent heat flux to the net radiative cooling in the cloud", so some, probably weak, influence of the heating on the transition is expected. This will be explored in more detail in a future paper.

## 8   Effects of biomass burning aerosol on precipitation

In our simulations, there is a clear mechanism in the autoconversion parameterisation for increased cloud droplet number concentrations to suppress precipitation, discussed earlier. Figure 14 shows the difference in rain rates and liquid water path between simulations with and without biomass burning aerosol. At the beginning of the polluted period, with both microphysics parameterisations we observe a large increase in LWP when smoke is present. This is interpreted in the next sections as mainly due to a suppression of entrainment of dry air from above the inversion. When smoke is added, the increased LWP makes more

difference than the increased CDNC, and so smoke aerosol increases the rain rate. The relative increase is slightly larger (over a factor of two on 7 Auguust) for the Khairoutdinov and Kogan (2000) scheme than the Kogan (2013) scheme, presumably because the compensating effect of higher CDNC is more important in the Kogan (2013) scheme. Towards the end of the smoke episode, when the smoke is more completely mixed into the boundary layer, as shown in Figure 6, the liquid water path



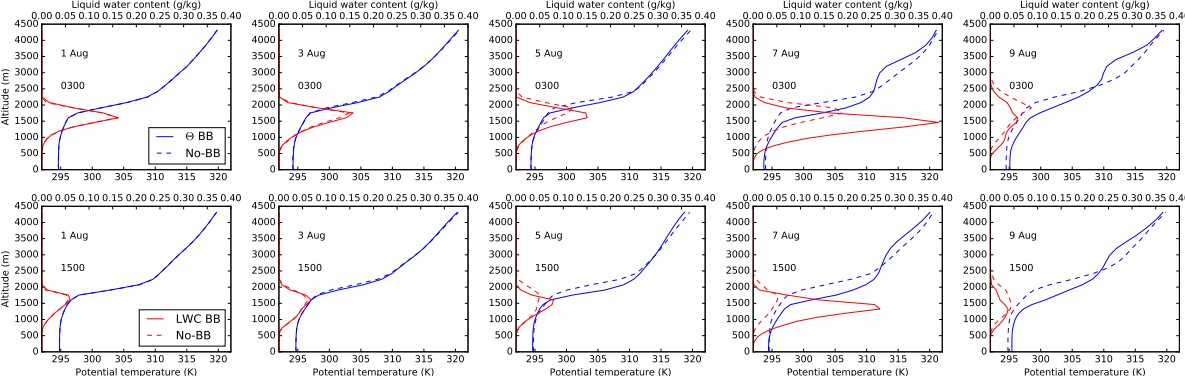

**Figure 13.** Domain-mean vertical profiles of potential temperature Θ and liquid water content (LWC) with and without biomass burning aerosol on odd numbered days at the times of day marked.

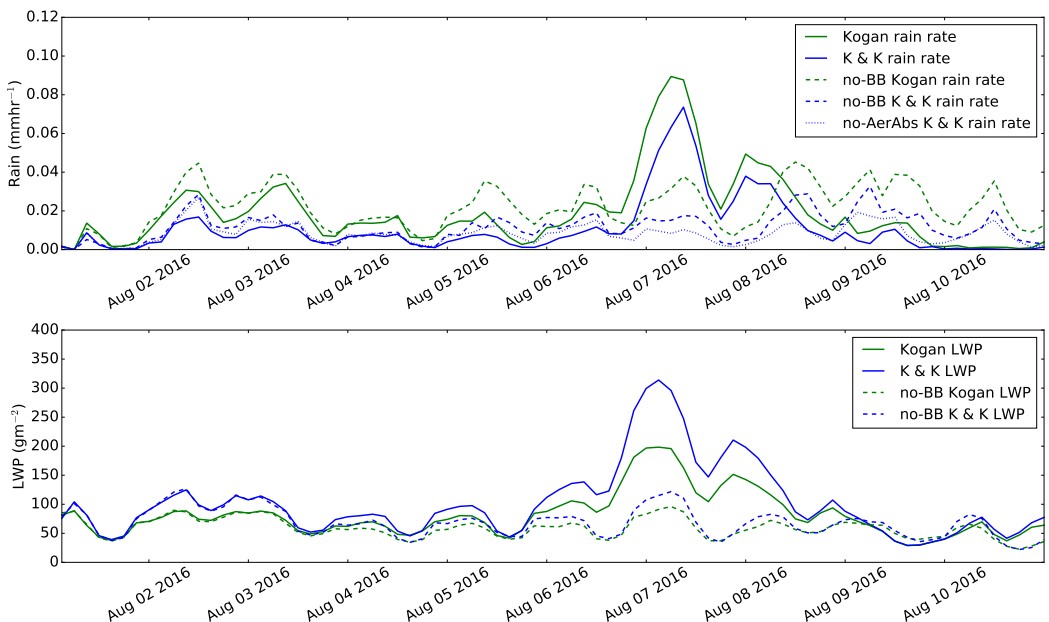

**Figure 14.** Domain mean precipitation rates (top) and liquid water path (bottom) in the regional model with and without biomass burning emissions, with the alternative parameterisations of autoconversion and accretion from Kogan (2013) and from Khairoutdinov and Kogan (2000) (labelled K & K).



**Table 2.** Selected mean rain rates, liquid water paths and fractional cloud cover over the clean and polluted periods of the simulation. The values tabulated are for the global model and then for the regional model with the alternative parameterisations of autoconversion and accretion from Kogan (2013) and from Khairoutdinov and Kogan (2000) (labelled K & K). We present simulations with and without biomass burning emissions ('BB' vs. 'no-BB'), and without aerosol absorption ('no-abs') for both microphysical parameterisations.

| Simulation | Glob no-BB | Glob no-abs | Glob default | K & K no-BB | K & K no-abs | K & K default | Kogan no-BB | Kogan no-abs | Kogan default |
|---|---|---|---|---|---|---|---|---|---|
| 1-5 Aug: Rain (mm hr$^{-1}$) | 0.007 | 0.006 | 0.005 | 0.09 | 0.008 | 0.006 | 0.019 | 0.016 | 0.014 |
| 1-5 Aug: LWP (g m$^{-2}$) | 92.4 | 96.4 | 106 | 73.9 | 73.9 | 81.1 | 61.9 | 63.0 | 67.8 |
| 1-5 Aug: CF (%) | 54 | 56 | 60 | 51 | 51 | 54 | 50 | 50 | 52 |
| 6-10 Aug: Rain (mm hr$^{-1}$) | 0.016 | 0.015 | 0.015 | 0.014 | 0.009 | 0.016 | 0.025 | 0.016 | 0.024 |
| 6-10 Aug: LWP (g m$^{-2}$) | 84.9 | 89.3 | 120 | 63.1 | 62.3 | 123 | 55.9 | 56.3 | 93.7 |
| 6-10 Aug: CF (%) | 44 | 43 | 62 | 44 | 43 | 63 | 44 | 43 | 60 |

is less strongly affected by the smoke (Figure 14) and the suppression of precipitation due to the increased CDNC is more important. The role of the two compensating effects is separated using the simulation with aerosol absorption switched off. For both microphysics schemes, Table 2 shows that in the polluted period the rain rates are reduced when non-absorbing smoke is introduced, then they increase again when the absorption is switched on and increases the LWP. The differences in the cloud

fraction between simulations are also summarised in this table, for discussion in Section 9. In the text that follows we first quote results from the more commonly used Khairoutdinov and Kogan (2000) scheme, but we also show that the choice of microphysical parameterisation affects the radiative effect of the aerosol.

## 9   Effects of biomass burning aerosol on radiation balance

### 9.1   Model results

The modelled outgoing shortwave radiation at the top of the atmosphere is compared to the CERES satellite on a domain mean basis in Figure 15. The trends in outgoing flux observed by the satellite are generally well captured by both regional and global models, but the outgoing flux is overestimated (by 23% on average in the regional model) compared to the satellite, mostly on 4, 7, 9, and 10 August. This suggests that the average cloud fraction and liquid water path (shown on the same plots as a red line, and discussed in Section 6 and Supplementary Figures S8 and S12) in the models is too high.

When biomass burning aerosol is added to a simulation without smoke emissions, the domain mean cloud droplet number and liquid water path both increase. The increase in liquid water path averages $60 \, \mathrm{g \, m^{-2}}$ in the last five days of the regional simulation, and $35 \, \mathrm{g \, m^{-2}}$ in the global model. This is larger than the $20 \, \mathrm{g \, m^{-2}}$ difference found in satellite data between low and high smoke cloud features by Wilcox (2010). The comparison may not be entirely fair because, for example, our aerosols are less separated from the cloud than is typical in the sample of Wilcox (2010) (from their Figure 1). The mean radiative effect

of the biomass burning aerosol over the second five days of the regional simulation is $-27.6 \, \mathrm{W \, m^{-2}}$ (a cooling); this compares



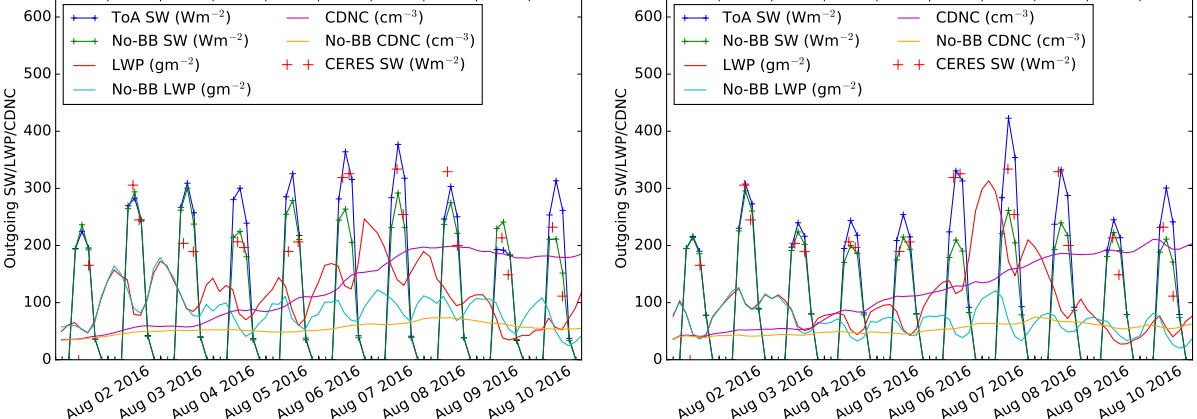

**Figure 15.** Outgoing shortwave flux at top of atmosphere ('ToA SW') in the baseline model and a model version without biomass burning emissions ('No-BB'), for the regional domain in the global model on the left and the regional model on the right. CERES domain-mean observations are shown on the figures in red crosses. The changes to CDNC and LWP across the 10 days are also shown.

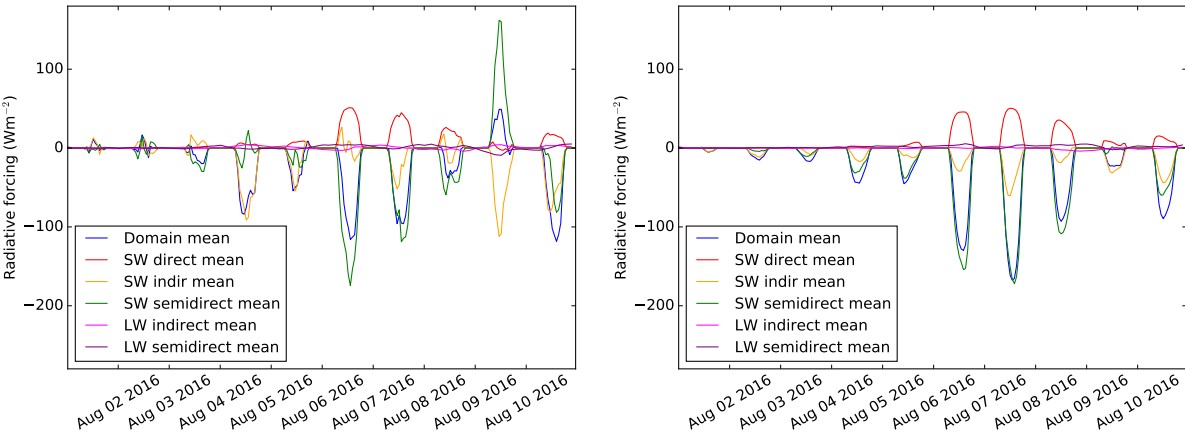

**Figure 16.** Decomposition of the radiative effects of biomass burning smoke in the regional model domain in the global model (left) and the regional model (right) into the direct, indirect and semidirect components in the short- and long-wave bands according to the recommendations by Ghan et al. (2012).

to $-6.9\,\mathrm{W\,m^{-2}}$ over the first five days. The corresponding increase in the aerosol burden compared to the simulation with no smoke emissions is 3.7. In the same area in the global model, the total radiative effect of the biomass burning aerosol is smaller than in the regional model: $-17.2\,\mathrm{W\,m^{-2}}$ over the second five days of the simulation. To explain the difference, we separate out the direct, indirect and semi-direct effects as follows.



**Table 3.** Radiative effects of biomass burning aerosol, in units $\mathrm{W\,m^{-2}}$. The values shown are averaged over the polluted period of the simulations, 6-10 August 2016. As in Table 2, we present values from the global model, and then from the regional models with the alternative parameterisations of autoconversion and accretion from Kogan (2013) and from Khairoutdinov and Kogan (2000) (labelled K & K).

| Radiative effect | Global K & K | Regional K & K | Regional Kogan |
|---|---|---|---|
| Direct SW | 10.3 | 11.4 | 9.67 |
| Indirect SW | -11.9 | -10.1 | -11.8 |
| Semi-direct SW | -17.0 | -30.5 | -23.0 |
| Indirect LW | 0.5 | -0.4 | -0.3 |
| Semi-direct LW | 0.9 | 2.3 | 1.8 |
| Total | -17.2 | -27.6 | -23.6 |

The method of Ghan et al. (2012) is used to isolate direct, semidirect and indirect radiative effects in Fig. 16. The same three simulations are compared as in Section 7: the default simulation, the simulation without biomass burning aerosol and the simulation without aerosol absorption. The short-wave direct effect is diagnosed as

$$\Delta S_{out,direct} = (S_{out} - S_{out,clean}) - (S_{out} - S_{out,clean})_{no-bb}, \tag{5}$$

the indirect effect (which is mainly due to changes in cloud droplet number and in liquid water path via precipitation suppression) as

$$\Delta S_{out,indirect} = S_{out,clean,no-aer-abs} - S_{out,clean,no-bb} \tag{6}$$

and the semi-direct effect (which is via changes in liquid water path due to heating effects) as

$$\Delta S_{out,semi} = (S_{out} - S_{out,no-bb}) - \Delta S_{out,direct} - \Delta S_{out,indirect} \tag{7}$$

In these equations, $S_{out}$ is the outgoing shortwave flux at the top of the atmosphere, $S_{out,clean}$ the outgoing shortwave flux without aerosols included in the calculation, and the other subscripts describe which simulation is used in each case. Strictly, Eq. 6 should read

$$\Delta S_{out,indirect} = S_{out,clean,no-aer-abs} - S_{out,no-bb,clean,no-aer-abs} \tag{8}$$

but we did not perform a separate simulation without either biomass burning or aerosol absorption, so we use the approximation that $S_{out,clean,no-bb} = S_{out,no-bb,clean,no-aer-abs}$. This is an assumption that the semi-direct effect from absorbing aerosol that is not from biomass burning is negligible. The radiative effects we find in our global and regional simulations are summarised in Table 3, and illustrated in Figure 17.

In the regional model, the direct effect is $+11.3\,\mathrm{W\,m^{-2}}$ over the polluted period, the indirect effect is $-10.1\,\mathrm{W\,m^{-2}}$, and the semi-direct effect is $-30.5\,\mathrm{W\,m^{-2}}$. The most striking result is the substantial semi-direct effect, driven by large increases





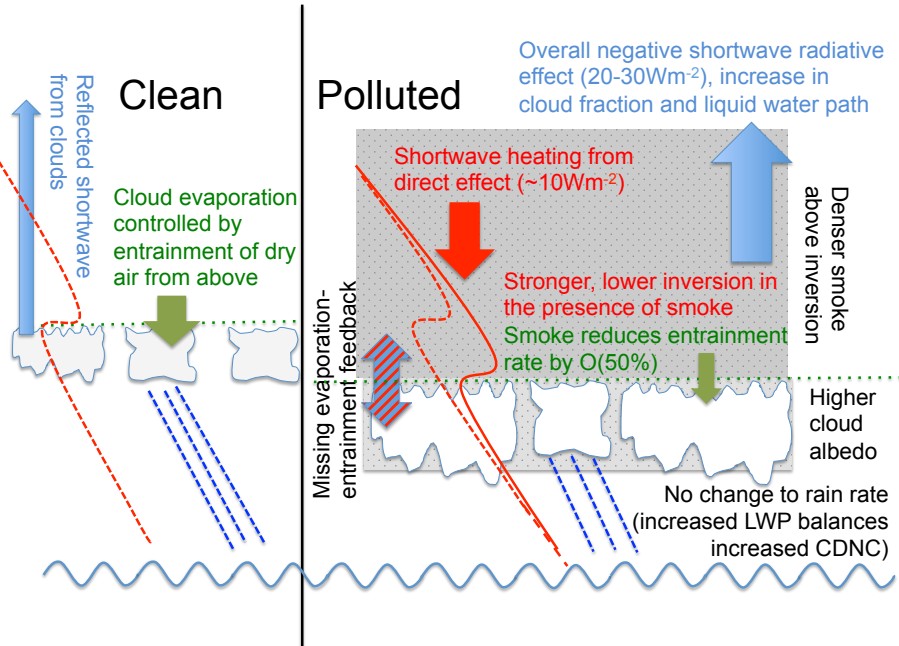

**Figure 17.** Diagram illustrating the different radiative effects of the smoke in our simulations

in liquid water path and fractional cloud cover. Qualitatively, this is expected from the observational studies of Wilcox (2012) and (Adebiyi and Zuidema, 2018) and we interpret it (below) in terms of changes to entrainment rates. The direct effect is also large, but it is only about a third of the magnitude of the total cloud forcing. This is not surprising in the middle of the Atlantic Ocean where the aerosol is mixing with the clouds and starting to be reduced by deposition. The direct effect is positive because the aerosol layer absorbs radiation that would otherwise be reflected by the clouds below it. Therefore, both the direct effect and the effect of increased droplet number on cloud albedo are likely to be overestimated due to the overestimated cloud fraction and LWP. The direct effect at an AOD of 1 was estimated by Chand et al. (2009) to increase by $0.86\,\mathrm{W\,m^{-2}}$ for each 1% increase in cloud cover. If our cloud fraction and LWP were in perfect agreement with observations, therefore, one would expect the radiative effects to be several $\mathrm{W\,m^{-2}}$ lower. The long-wave indirect effect is $-0.4\,\mathrm{W\,m^{-2}}$ and the long-wave semi-direct effect is $+2.3\,\mathrm{W\,m^{-2}}$, which we interpret as a small reduction in long-wave cooling at cloud top due to the biomass burning aerosol.

With the microphysical parameterisation of Kogan (2013) is used instead of that of Khairoutdinov and Kogan (2000), the overall forcing is smaller, at $-23.6\,\mathrm{W\,m^{-2}}$ when averaged over the domain and the last five days of the simulation. While the mean LWP is in good agreement with observations during this period, the outgoing shortwave flux is still biased high by 15% over the ten days, compared to 23% with the default microphysics scheme. The components of the aerosol radiative effect are plotted in Supplementary Figure S11. The long-wave radiative effects are similar to those with Khairoutdinov and Kogan (2000) scheme. The direct effect is slightly reduced to $+9.7\,\mathrm{W\,m^{-2}}$, due to the lower cloud cover. However, the indirect effect





increases slightly to $-11.8\,\mathrm{W\,m^{-2}}$, despite the lower cloud liquid water path, and the magnitude of the semi-direct effect decreases strongly to $-23.0\,\mathrm{W\,m^{-2}}$. From Table 2, aerosol absorption increases LWP by $61\,\mathrm{g\,m^{-2}}$ with the Khairoutdinov and Kogan (2000) scheme but $37\,\mathrm{g\,m^{-2}}$ with the Kogan (2013) scheme. A possible reason for this lower semi-direct effect with the Kogan (2013) scheme is that the entrainment rates across the boundary layer are slightly higher with the Kogan (2013)

scheme. The domain mean entrainment rate across the inversion over the polluted period with the Khairoutdinov and Kogan (2000) microphysics scheme in the regional model is $0.019\,\mathrm{m\,s^{-1}}$ with biomass burning aerosol and $0.042\,\mathrm{m\,s^{-1}}$ without. With the Kogan (2013) scheme, the entrainment rate is $0.022\,\mathrm{m\,s^{-1}}$ with biomass burning aerosol and $0.047\,\mathrm{m\,s^{-1}}$ without. In the absence of aerosol absorption, the cloud droplet number concentration is slightly higher than when absorption is switched

on (Supplementary Figure S10). With a reduced entrainment rate due to the heating effect of the aerosol layer, fewer cloud condensation nuclei are entrained into the clouds, or the mean updraft speed is reduced.

In the global model, the instantaneous short-wave direct and indirect effects were also calculated for the regional domain. Averaged over the polluted period, the direct effect is $+10.3\,\mathrm{W\,m^{-2}}$, the semi-direct effect is $-17.0\,\mathrm{W\,m^{-2}}$ and the indirect effect is $-11.9\,\mathrm{W\,m^{-2}}$. On average, the direct and indirect effects are in good agreement with the regional model with the same Khairoutdinov and Kogan (2000) microphysics scheme. However, the instantaneous radiative effects in the global model

tend to fluctuate more than in the regional model. This is probably mostly because the cloud fraction (Supplementary Figure S8) tends to more extreme values than in the regional model. Because the sub-grid cloud fraction is prognostic in the global model, the cloud fraction is more correlated between model grid-boxes, and therefore small fluctuations are less likely to be averaged out. In particular, the differences in cloud fraction between the simulation without smoke and the simulation without aerosol absorption are larger in the global model than the regional model. On 9 August, this leads to large opposing indirect and semi-

direct effects, which mostly cancel overall (Figure 16) but still lead to a positive radiative effect not observed in the regional model. This appears to be responsible for the difference in aerosol radiative effects between regional and global models when averaged over the polluted period (Table 3), and highlights the sensivity of aerosol radiative effects to the representation of sub-grid cloud in models.

The different average semi-direct radiative effects of the smoke in the two models can be at least partly explained by different

entrainment rates. In the global model, the entrainment rate is reduced from $0.023\,\mathrm{m\,s^{-1}}$ to $0.013\,\mathrm{m\,s^{-1}}$ when smoke is added, while in the regional model the reduction is from $0.042\,\mathrm{m\,s^{-1}}$ to $0.019\,\mathrm{m\,s^{-1}}$. The smaller entrainment rate, and smaller effect of smoke on entrainment, in the global model compared to the regional model is presumably due to the different vertical or horizontal resolution.

## 9.2   Comparison with other studies

The increase in LWP in the presence of smoke in our regional model with the Khairoutdinov and Kogan (2000) microphysics parameterisation seems to be mostly due to a strengthened inversion. The stronger inversion reduces entrainment of dry air from the free troposphere into the clouds, leading to reduced evaporation. There is a consensus in the literature that aerosol mostly above stratocumulus clouds, as in our case study, tends to make inversions stronger, but aerosol within clouds tends to weaken them (Johnson et al., 2004; Koch and Del Genio, 2010; Zhou et al., 2017).





The widely varying changes in cloud liquid water path observed by other studies in the presence of smoke aerosol can be at least partly explained by the degree with which the smoke mixes into the clouds. The LES simulations of Yamaguchi et al. (2015) suggested the LWP should increase in the presence of partly entrained smoke aerosol, while those of Zhou et al. (2017) found a decrease. Zhou et al. (2017) presented evidence that their decreasing LWP is controlled by sedimentation-evaporation-entrainment feedbacks (see Section 2), which are not represented in our model. In their default simulation, the smoke layer descends into the clouds almost in its entirety, with much higher concentrations of absorbing aerosol below than above the boundary layer by 60 hours into their simulation (their Figure 3). The authors also performed a simulation with the base of the smoke layer initially located 400 m higher. This leads to less complete mixing, which may be more similar to our case study and to that of Yamaguchi et al. (2015) (see our Figure 6). In this second study with higher-altitude smoke, Zhou et al. (2017) found the LWP increased relative to the simulation with more complete mixing, but still decreased relative to the simulation without smoke aerosol. We aim to include the sedimentation-evaporation-entrainment feedback in subsequent studies to see if we can reproduce this LES result.

While Lu et al. (2018) do not separate direct and semi-direct radiative effects in their regional WRF simulations, and the effects have opposite signs so are not constrained to be small, we can speculate, based on the very small changes in LWP between their 'M' and 'P' (default) simulations, that their time-averaged semi-direct effect must be substantially smaller than that in our regional model for the last five days of August 2016. This is confirmed by the increase in cloud top height and cloud top entrainment found by Lu et al. (2018) in the presence of biomass burning, opposite to our result. They also average over pristine periods without much smoke aerosol, while we quote results only in a very polluted period.

Similar to the LES study of Zhou et al. (2017), the analysis of satellite observations by Costantino and Bréon (2013) found a negative correlation of liquid water path with AI when aerosol is mixed into clouds in this region. However, they also found a (weak) positive correlation of cloud fraction with AI. As in the LES studies, the authors interpret their results in terms of entrainment of dry air: well-mixed aerosol leads to a weaker inversion with more entrainment, and this leads to more evaporation of clouds.

Our indirect radiative effects can also be compared to the 'remote' results in the WRF regional modelling study of Lu et al. (2018). When averaging over a two-month period, the authors find a time-averaged indirect effect of $-6.12\,\mathrm{W\,m^{-2}}$ in their remote region, compared to our estimate of around $-10\,\mathrm{W\,m^{-2}}$ using the Khairoutdinov and Kogan (2000) microphysics parameterisation. Lu et al. (2018) simulate a mean increase in CDNC of a factor of two from around the same background level as in our simulations ($50\,\mathrm{cm^{-3}}$), while we simulate an increase of a factor 3. The mean increase in LWP they find is 9% (where these figures are presented for the whole SE Atlantic region in daytime, and we assume they are similar to those for the remote region). Without absorbing aerosol in their simulations (their 'M' case) they find a similar increase in LWP, implying that the increase they observe is mainly due to precipitation suppression by aerosol, hence the indirect effect, rather than to the semi-direct effect. Comparing our simulations without smoke and without aerosol absorption, we find no significant change in LWP due to precipitation suppression with the Khairoutdinov and Kogan (2000) microphysics parameterisation, but a small increase of 5% using the Kogan (2013) scheme, from Table 2. The comparison would be fairer if the cloud fraction was the




same, but in the last five days of our simulations for August 2016 it appears the cloud fraction is higher, based on a visual inspection of Figure S2 in Lu et al. (2018).

## 10 Conclusions

The region in which biomass burning aerosol interacts with South Atlantic clouds is studied in the HadGEM climate model with prognostic aerosol number concentration from the UKCA aerosol microphysics module, in regional and global configurations. Biomass burning aerosol increases the total aerosol burden by a factor of four and the cloud droplet numbers by a factor of three to a few hundred per cubic centimetre, and has substantial effects on the cloud properties. In our regional simulation, a square of length $1200\,\mathrm{km}$ centred near Ascension Island, resultant large radiative effects sum to $-27.6\,\mathrm{W\,m^{-2}}$. The aerosol also has quite large effects on dynamics and microphysics, reducing the height of the inversion by up to $200\,\mathrm{m}$. Increased cloud droplet nuber concentration suppresses rain, though in our simulations this is offset by dynamically induced increases to the cloud liquid water path. The large increases in liquid water path we observe result from a strong suppression of the entrainment rate when smoke aerosol is added. The aerosol plume we studied had substantial direct radiative effects, averaging $+10\,\mathrm{W\,m^{-2}}$ in the regional simulation, and these partly offset the cloud forcing of around $-40\,\mathrm{W\,m^{-2}}$. In our simulations, increased CCN concentrations lead to strong cooling. Less intuitively, increased absorption of solar radiation can also cool this marine region via cloud adjustments, as noted by Koch and Del Genio (2010) and Sakaeda et al. (2011).

Our simulations highlight some areas where further work is needed to ensure good agreement between the model and observations, which will be facilitated by recent field measurements in this region. First, the transport of aerosol at large scales is imperfect: the aerosol layers are too low. This also leads to discrepancies in the relative humidity at high altitude, and this may affect the boundary-layer clouds via entrainment. If more aerosol mixes into the boundary layer because it is lower in altitude in our simulations than in reality, the effects on clouds may be exaggerated. Second, our treatment of aerosol activation reproduces the observations of CDNC we have used here reasonably well, but it might be right for the wrong reasons (as the hygroscopicity of the aerosol and the updraft speed are tuned), and should be studied further. Third, we showed that the means by which aerosol is depleted, in rain, is sensitive to the parameterisation of autoconversion and accretion in the microphysics scheme. The choice of parameterisation also strongly influences the components of the radiative effect of aerosol. Further work might indicate more conclusively which scheme is most appropriate. Fourth, we do not simulate the dependences of entrainment on cloud droplet number concentration via sedimentation and evaporation. Last, the depletion of aerosol by wet scavenging was recently shown to be important to the stratocumulus-to-cumulus transition, at least in some situations (Yamaguchi et al., 2017). Removal of aerosol by drizzle hastens the onset of more drizzle by increasing the collision-coalescence rate, and the drizzle reduces the liquid water path, hence the cloud cover. Both these last issues, but in particular the sedimentation feedback, which is not represented in our model, highlight the need for fully prognostic cloud droplet number concentrations. Once these are included and the coupling between aerosol and cloud microphysics is improved, we should be better able to reproduce the short-lived depletions of CCN seen in the surface observations (Figure 6).



As the simulations are improved and examined in more detail, the benefits of model resolutions intermediate between those of general circulation models and those of large eddy models will either become more apparent, or it will become clearer that they don't help. We showed here that a 4 km model produces more realistic variability in the boundary-layer CCN concentrations than the global model, but still not as much variability as in observations. This might be improved by better coupling of the aerosol and cloud microphysics schemes in our model, or by higher resolution simulations. If the total aerosol concentration depends on the model resolution in these subsequent studies, this would indicate that non-linear processes are important and higher resolution is needed in general circulation models to resolve them. If, on the other hand, the regional averages are consistent between simulations at different resolutions, then higher resolution may not be as important.

Despite its shortcomings, the performance of our model is in many respects very good. The global and regional models are more-or-less consistent in terms of the mean and spatio-temporal changes in aerosol number concentrations, CDNC, and LWP, giving confidence in the treatment of the boundary conditions. The cloud top and inversion heights are generally in good agreement with observations both for regional and global models, although the high sensitivity of radiative effects to these dynamical properties suggests further work is warranted. Trends in the cloud fraction and LWP match the observations well, although the LWP is overestimated, and since the outgoing SW fluxes also track the CERES satellite, there is a successful closure that gives confidence in the model physics. Despite an overestimate of the absolute magnitude, the timeseries of precipitation appears to be relatively well represented too. The regional model captures spatial variation in CDNC and temporal variation in liquid water path better than the global model.

The good performance of the regional model suggests it is ready for a second level of evaluation, using the detailed in-situ measurements from the CLARIFY campaign. It also motivates the study of the stratocumulus to cumulus transition in this region with the model, especially once fully double-moment cloud microphysics is incorporated and compared with the current setup. We hope that this first model evaluation study, and future simulation efforts that build on it, will facilitate continued efforts to fully exploit the plethora of new observation data in this area.

## 11  Acknowledgements

This research was funded by the NERC CLARIFY project NE/L013479/1. AMSR data are produced by Remote Sensing Systems and were sponsored by the NASA AMSR-E Science Team and the NASA Earth Science MEaSUREs Program. Data are available at www.remss.com. The development of the CPP algorithm was done by Hartwig Deneke, Jan Fokke Meirink, Rob Roebeling, and Erwin Wolters (KNMI). Cloud condensation nuclei and radiosonde data were obtained from the Atmospheric Radiation Measurement (ARM) Climate Research Facility, a U.S. Department of Energy Office of Science user facility sponsored by the Office of Biological and Environmental Research. The GPM precipitation data were provided by the NASA Goddard Space Flight Center's GPM team and PPS, which develop and compute the IMERG dataset as a contribution to GPM, and archived at the NASA EarthData archive. We also thank Jim Haywood and the ARM site team for providing rainfall rates at Ascension Island. We acknowledge use of the Monsoon2 system, a collaborative facility supplied under the Joint Weather and Climate Research Programme, a strategic partnership between the UK Met Office and the Natural Environment Research



Council (NERC). We also used the JASMIN facility (www.jasmin.ac.uk/) via the Centre for Environmental Data Analysis, funded by NERC and the UK Space Agency and delivered by the Science and Technology Facilities Council. Simulation data are available from this facility upon reasonable request.



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
