# Peer review of "Large simulated radiative effects of smoke in the south-east Atlantic"

_Atmospheric Chemistry and Physics, 2018_

## Referee Comment (RC1) · Anonymous Referee #1 · 29 Jun 2018

In this study, Gordon et al. studied the radiative effects of smoke induced over a 1200 km2 area of south-east Atlantic using HadGEM climate model at convection-permitting and global resolution in conjunction with several satellite observations and ground based observations (from LASIC campaign over the Ascension Island). The study finds that smoke aerosols can induce significant cooling effect over this region via alter cloud properties. Whether the semi-direct effect or the indirect effect dominates depends on the selection of autoconversion scheme. The paper has significant scientific merit and is clearly written with great details. The reviewer recommends publication after some minor revisions.

General comments:

1. As for regional-to-global scale modeling study, 10-day period seems to be too short

for estimating radiative effects of smoke. As shown in Figure 1a of Zuidema et al. (2018, GRL), during the fire season over this region, the fire activity fluctuates widely from day to day. By the end of the simulation period (i.e. Aug. 8), the smoke concentration as measured over Ascension Island was much higher than any other days (except Aug. 13) in 2016. Therefore, -22.0Wm-2 cooling sounds very dramatic but the authors failed to emphasize that this value is calculated for the most polluted period during the fire season in many places. The authors can either run model (global model only if regional model is too expensive) for longer period, or emphasize the period and region, over which the model simulation is conducted in text, especially in the abstract.

2. Although the results in the paper are thoroughly discussed with many details, the reviewer suggests the authors to include some additional results for the interests of the scientific community. Firstly, in addition to show the time series of modeled and observed AODs in Figure 8, please include the spatial patterns of modeled AOD in comparison with MODIS observations (averaged over study period or during Aug. 2 and Aug. 7) in main text or supplementary. By doing so, the authors can further justify the performance of HadGEM/UK Met Office Unified models. Secondly, many previous modeling and observational studies disagree on the effect of smoke on cloudtop height. The reviewer strongly suggests the authors to examine smoke-induced changes in cloud-top heights by comparing three cases after the discussion of Figure 13.

Specific comments:

The abstract seems to be way too long – almost close to 500 words. Please condense the abstract, if possible around 300 words.

Page 2, line 12: You mean" Once entrained, smoke. . .can have very different effects to cloud aloft"?

Page 2, line 15: This hypothesis may be true, but please explain why this is the case. (For example, maybe cite Figure 3 in Chand et al., 2009, Nature geoscience?)

Page 3, Figure 1: It may be better to label Ascension Island in the figure instead of saying it in the caption.

Page 4, line 5 to line 8: I am not familiar with the models used in this study. To drive a model with 4km resolution with a model with 65 km resolution, is this the downscaling or nesting technique? For regional models like MM5 and WRF, the ratio between the resolutions of the outer domain and inner domain is usually 3:1 or 5:1. As in this study, the ratio is 16:1, which is quite large. With such large ratio, can signature of smoke plumes in meso-scale be properly simulated?

Page 5, line 5 to line 7: please rewrite this sentence, using the phrase like "the vertical resolution of model grid boxes..."

Page 6, line 4 and line 14: 120nm is typical size of freshly emitted size of smoke particles; however, it should be in accumulation mode. Why put smoke particles into insoluble Aitken mode?

Page 6, line 4: "...highest at the surface...reach zero at 3km above ground." Is this based on any observation or modeling studies? If so, please cite the reference.

Page 6, line 10:"In any case,...rather than emission" Is this the case? Is this based on any studies? If so, please cite the reference.

Page 6, line17: The authors only mentioned BC, how about POM emissions? What is the mass ratio between BC/POM in this study?

Page 8, line 26: Since the second aerosol indirect effect is considered in the case with Kogan (2013) scheme, does that mean this case is more realistic compared to the case with default scheme?

Page 10, line 10: Change "8th August" to "8 August " to be consistent with the text.

Page 13, line 18: Change "height of the aerosol layer" to "the height of the aerosol layer bottom".

Page 17, line 2: Any reason such large box is used?

Page 19, line 17: If I remember correctly, high humidity associated with smoke plume as mentioned in (Adebiyi et al., 2015) is due to water vapor emissions from fires. Are water vapor emissions considered in the study?

Page 28, line 2: By "table" you mean table 2?

Page 28, line2-line5: It is very confusing when the authors say "compare line x to line y". In addition, I believe that the line numbers are not correct according to Table 2. Please rewrite the text in parentheses.

Page 33, section 9.3: It would be better to compare modeled semi-direct effect to Sakaeda et al. (2011) in this section.

---

## Referee Comment (RC2) · A. Ackerman (Referee) · 2 Aug 2018

Overall

This well written study of the impacts of biomass-burning aerosol (BBA) on low-level clouds over the SE Atlantic covers a lot of ground without over reaching. The simulations are clearly described, the analysis solid, and results fairly put in the context of previous studies. It is publishable after addressing a few minor points detailed below, in order of appearance.

Specific comments

1. Page 5: if winds are being nudged, would that not diminish any dynamic response of large-scale vertical motion to a heating perturbation?

[Figure]

2. Surprising that changing sigma_w from eqn. 1 to just fixing it at 0.12 m/s would have much of an effect in increasing CDNC, given that eqn. 1 already includes a floor of 0.10 m/s. Also, why not just increase the floor to 0.12 m/s in the original expression?

3. The kappa value for sulphate seems quite large. Typical values for ammonium bisulphate are more like 0.6, maybe 0.7 at the extreme (cf. Petters and Kreidenweiss 2007). Using a value of effectively unity seems worth an explanation.

4. What is basis for fraction of each grid box that is raining being 0.3 on the 65 km mesh?

5. Interquartile range is a scalar, and what is shaded in fig. 3 is the inner half of the distribution.

6. Might point out in the last paragraph of section 6.1 that while the moisture aloft is indeed associated with BBA aloft over the region, they are associated with a common factor, namely outflow from the deep continental boundary layer, and surely the moisture would be there even in the absence of aerosol sources.

7. The generally good agreement between aerosol index AI and aerosol optical depth AOD described in the first full paragraph of page 19 does not get so much support from fig. 8, which instead shows that 2 of the 5 so-called clean days have quite large values of AI but not AOD. Not sure I'd characterize the relationship as generally good and wonder how robustly they are correlated.

8. MODIS platforms pass overhead not far from local noon. Do the comparisons with model result in fig. 10 use the same, quite limited sampling of the diurnal cycle?

9. The positive LW semi-direct forcing is confusing, given that lower cloud tops for polluted clouds should emit more LW. Less emission from even warmer surface through more overcast deck, perhaps?

10. The entrainment rates discussed on page 33 seem higher than expected, with values of around 2 to 5 cm/s. Not hard to find literature reporting typical rates for

nocturnal stratocumulus 5 if not 10 times smaller, even smaller during the day. Is mean entrainment rate computed from the mean of column-wise $d(z_i)/dt - w(z_i)$? The plausible simulation of PBL depth does not obviously square with the reported rates.

Typesetting issues

1. Line 6 of page 3 should use \citet instead of \citep.

2. Year for Gunnar et al. citation is missing on line 12 of page 6.

6. The citation D. Holdridge...2016a appears in multiple places but is missing from the references.

---

## Author Response (AR2)

**Review replies to Large simulated radiative effects of smoke in the south-east Atlantic**

Hamish Gordon[1], Paul R. Field[1,2], Steven J. Abel[2], Mohit Dalvi[2], Daniel P. Grosvenor[1], Adrian A. Hill[2], Ben T. Johnson[2], Annette K. Miltenberger[1], Masaru Yoshioka[1], and Ken S. Carslaw[1]

[1]School of Earth and Environment, University of Leeds, LS2 9JT, United Kingdom
[2]Met Office, Fitzroy Road, Exeter, EX1 3PB, United Kingdom

*Correspondence to:* Hamish Gordon hamish.gordon@cern.ch

We thank both reviewers for their thoughtful reading of the manuscript and helpful comments. We feel that through their suggestions we have been able to significantly improve our paper.

**1 Reviewer 1**

**1.1 General comments**

5   **As for regional-to-global scale modeling study, 10-day period seems to be too short for estimating radiative effects of smoke. As shown in Figure 1a of Zuidema et al. (2018, GRL), during the fire season over this region, the fire activity fluctuates widely from day to day. By the end of the simulation period (i.e. Aug. 8), the smoke con- centration as measured over Ascension Island was much higher than any other days (except Aug. 13) in 2016. Therefore, -22.0Wm-2 cooling sounds very dramatic but the authors failed to emphasize that this value is calculated for the most polluted**

10   **period during the fire season in many places. The authors can either run model (global model only if regional model is too expensive) for longer period, or emphasize the period and region, over which the model simulation is conducted in text, especially in the abstract.**

We only became aware of the 2018 Zuidema paper during our review process, and indeed this paper shows that we picked (more or less by accident) the start of the period with the most smoke over Ascension Island of any episode in the 2016 fire

15   season. Rather than running longer simulations, which would entail a great deal more analysis, we prefer to emphasise in the text that the period we have chosen is not representative of the whole fire season.

We added a sentence to the abstract: *At Ascension Island, this smoke episode was the strongest of the 2016 fire season.* To the introduction, we add *Our test case runs from 1 to 10 August 2016. The first five days of this period were approximately representative of clean conditions in the mid-Atlantic, while the second five days cover the first half of the largest smoke*

20   *plume to reach Ascension Island in the 2016 fire season. The single-particle soot photometer on Ascension recorded a peak in refractory black carbon concentration of $1.5\mu gm^{-3}$ on 9 August, while the monthly mean for August is $0.5\mu gm^{-3}$ (Zuidema et al , 2018).* When we compare to other studies, we say, with respect to Lu et al (2018), *They also average over pristine periods without much smoke aerosol, while we quote results only in a very polluted period.* We also added a note to the caption of

the table presenting our results on radiation balance: *We emphasise that the results are averaged over one of the most polluted periods in the 2016 fire season.* Finally, we added a qualifier in the conclusions as follows: *resultant large radiative effects sum to* $-27.6 \, \mathrm{W \, m^{-2}}$*, in an especially strong smoke episode.*

**Although the results in the paper are thoroughly discussed with many details, the reviewer suggests the authors to include some additional results for the interests of the scientific community. Firstly, in addition to show the time series of modeled and observed AODs in Figure 8, please include the spatial patterns of modeled AOD in comparison with MODIS observations (averaged over study period or during Aug. 2 and Aug. 7) in main text or supplementary. By doing so, the authors can further justify the performance of HadGEM/UK Met Office Unified models.**

We are happy to provide the spatial patterns of AOD (and, in response to reviewer 2, aerosol index) and compare the AOD to the model. We added a new supplementary figure showing AOD and AI on 1,3,5,7,9 August. We added the following to the text: *Despite the better correlation of aerosol number concentration expected with AI than AOD, the AI increases before the AOD while the CCN at Ascension Island increases after. This is likely due to the vertical mixing of the aerosol layer. The layer is initially at high altitude, which leads to a higher AI than lower-lying aerosol layers would (Herman et al, 1997) but does not affect AOD. Then, as the layer thickens, the AOD increases more quickly than the AI. The spatial patterns of AI, retrieved AOD and simulated AOD are shown in Supplementary Figure S8.*

**Secondly, many previous modeling and observational studies disagree on the effect of smoke on cloud- top height. The reviewer strongly suggests the authors to examine smoke-induced changes in cloud-top heights by comparing three cases after the discussion of Figure 13.** Indeed, in our short simulation we find smoke tends to suppress cloud top height, in line with observations, while Lu et al (2018) and Sakaeda et al (2011) find the opposite. The effect we see is mainly nocturnal. Sakaeda et al atttribute their findings to reductions in near-surface temperature (which we, and Lu et al, are not sensitive to, with fixed SST), increases in free-tropospheric temperature, and reduced large-scale subsidence. We commented on this in the text that was published in ACPD - but this was an update from the originally submitted version, and we don't think the reviewer saw it: *As observed by Wilcox (2010), the cloud also gets thicker, as the cloud top rises while the cloud base height decreases. However, the WRF model of Lu et al (2018) produces an increase in cloud top height in the presence of biomass burning aerosol, and the authors attribute the observed reductions to co-variability of aerosols and meteorology.* Lu et al state in their supplementary that 50% of the cloud top height difference they see is due to enhanced entrainment and 50% due to reduced subsidence. On average, we also find smoke reduces subsidence (by a mean of 6.8% in the polluted period in the first 9km of altitude), but in our simulations it also reduces entrainment. We added some numbers to the text in our previous iteration. There are reasons why our simulations might differ to Lu et al: they can simulate faster evaporation of smaller water droplets at cloud top, but we can't, because our cloud microphysics scheme only sees cloud droplet number concentrations in its autoconversion and accretion parameterizations. It seems unlikely that this effect could be strong enough to reverse changes in cloud top height, but it may contribute. It is also likely that these effects depend strongly on the height of the aerosol layer. We added to the text *The reduction in entrainment rates due to smoke we observe is in contrast to the increased entrainment rate seen by Lu et al (2018), and may help to explain why on average we see a reduction in cloud top height due to smoke while they see an increase.*

In our simulations, unsurprisingly, changes in potential temperature and liquid water content we see are entirely due to the absorption effect of smoke, rather than any microphysical effect. To demonstrate this we added the profiles of potential temperature and liquid water content in the simulation without aerosol absorption to Figure 13. We now also provide the same plot for the global model as a Supplementary Figure. The results from the global model are qualitatively consistent, but more structures are seen, probably because there are fewer grid boxes to average over. We added some more discussion of the figure. *The simulation without aerosol absorption tracks the simulation without smoke, so the changes observed are due to the absorbing properties of the smoke. The regional model is qualitatively consistent with the global model (Supplementary Figure S12).*

**1.2 Specific comments**

– **The abstract seems to be way too long – almost close to 500 words. Please condense the abstract, if possible around 300 words.** We shortened the abstract to below 350 words.

– **Page 2, line 12 'Once entrained, smoke can have very different effects to cloud aloft'.** We did mean 'smoke' here. In other words, the altitude of the heating effect due to smoke is important - if smoke is above the boundary layer we expect it to reduce the inversion height, while if smoke is in the boundary layer we expect it to increase the inversion height.

– **Page 2, line 15: This hypothesis may be true, but please explain why this is the case. (For example, maybe cite Figure 3 in Chand et al., 2009, Nature geoscience?)** We now mention Figure 2 in Chand et al: *This is broadly consistent with Figure 2 in Chand et al (2009). Smoke tends to lie above clouds east of this line, while large scale subsidence usually leads to more frequent mixing of smoke into clouds to the west.*

– **Page 3, Figure 1: It may be better to label Ascension Island in the figure instead of saying it in the caption.** Done

– **Page 4, line 5 to line 8: I am not familiar with the models used in this study. To drive a model with 4km resolution with a model with 65 km resolution, is this the downscaling or nesting technique? For regional models like MM5 and WRF, the ratio between the resolutions of the outer domain and inner domain is usually 3:1 or 5:1. As in this study, the ratio is 16:1, which is quite large. With such large ratio, can signature of smoke plumes in meso-scale be properly simulated?** We have no technical problems with using a 16:1 ratio for the regional to global models. This is commonly used in regional modelling with the UM, and we could go to even higher ratios, for example Grosvenor et al (2017) use a 25 km-resolution global model and a 1 km resolution regional model. Close to the boundaries of the domain, there is not much difference from one 4km grid-box to the next, while in the interior this is not the case. In our analysis we provide means from the 200x200 grid boxes in the centre of the domain, rather than the 300x300 grid boxes of the full domain, in order to account for the transition to regional resolution. A smaller ratio of resolutions would reduce the distance into the regional model that is needed for this transition to the regional resolution, so it might be a bit more efficient for large domains.

- **Page 5, line 5 to line 7: please rewrite this sentence, using the phrase like "the vertical resolution of model grid boxes"** We added *with the spacing increasing as the altitude increases* to clarify this.

- **Page 6, line 4 and line 14: 120nm is typical size of freshly emitted size of smoke particles; however, it should be in accumulation mode. Why put smoke particles into insoluble Aitken mode?** The insoluble Aitken mode is named as in the model code. We have only one insoluble mode, but it does not have a fixed radius and mass within it is not transferred to other modes if its radius gets too large (as happens in the soluble modes). Instead mass is transferred to the soluble modes as the aerosol ages, and by the time the smoke aerosol reaches our regional model domain, it is almost entirely in the soluble accumulation mode.

  So we could have called this Aitken insoluble mode an accumulation mode, but for the sake of readers accustomed to our model we prefer to stick to the convention of calling it an Aitken mode. The reason for the convention is that the mode radius can be influenced by fossil fuel and biofuel emissions (which are assigned a diameter of 60nm, in accordance with AeroCom recommendations) as well as biomass burning emissions at 120nm.

- **Page 6, line 4: "…highest at the surface…reach zero at 3km above ground." Is this based on any observation or modeling studies? If so, please cite the reference** We did not distinguish between grassland fires, which are best simulated with emissions at the surface, and tropical forest fires, which are best simulated in our model with uniform emissions from the surface to 3km altitude, according to Johnson et al (2016). The FEER inventory is based on satellite data so has no vertical emissions information. Therefore we compromised and assumed a tapering profile from the surface to 3km. We added an explanation of this to the manuscript text. We tried several choices and found it mades very little difference to aerosol concentrations in our remote region. The advantage of the FEER inventory compared to the GFED climatology is the higher spatial and temporal resolution, which is likely to make more difference than the vertical emissions profiles.

- **Page 6, line 10:"In any case,…rather than emission" Is this the case? Is this based on any studies? If so, please cite the reference.** For "In any case,…" we fixed the citation Gunnar et al to read Myhre et al 2003.

- **Page 6, line17: The authors only mentioned BC, how about POM emissions? What is the mass ratio between BC/POM in this study?** We agree this is valuable information to add. We now state that *After ageing, in our simulation domain we find a ratio of black carbon to organic carbon of approximately 1:11, which is within the range of observations in the literature, e.g. in Table 1 and Table 2 of Formenti et al (2003).*

- **Page 8, line 26: Since the second aerosol indirect effect is considered in the case with Kogan (2013) scheme, does that mean this case is more realistic compared to the case with default scheme?** The second indirect effect of aerosols is considered in both Khairoutdinov & Kogan and Kogan schemes, it is just stronger in the case of the Kogan scheme.

- **Page 10, line 10: Change "8th August" to "8 August " to be consistent with the text.** Done

– **Page 13, line 18: Change "height of the aerosol layer" to "the height of the aerosol layer bottom"** We changed to "the height of the peak of the aerosol layer".

– **Page 17, line 2: Any reason such large box is used?** We changed the large box to be instead our usual average over the centre of the simulation domain.

– **Page 19, line 17: If I remember correctly, high humidity associated with smoke plume as mentioned in (Adebiyi et al., 2015) is due to water vapor emissions from fires. Are water vapor emissions considered in the study?** Water vapor emissions from fires are not considered, and we added a clause to the text to specify this. Based on other analysis we have done outside the scope of this paper, we don't think water vapor emissions from fires are as important as convective transport of water to higher altitudes close to the coastline of Africa. See also the comment of reviewer 2.

– **Page 28, line 2: By "table" you mean table 2?** Yes, we do mean table 2. Thanks!

– **Page 28, line2-line5: It is very confusing when the authors say "compare line x to line y". In addition, I believe that the line numbers are not correct according to Table 2. Please rewrite the text in parentheses**. We had corrected the references to lines in the table in the version uploaded to ACPD after our first initial submission. Sorry for the confusion.

– **Page 33, section 9.3: It would be better to compare modeled semi-direct effect to Sakaeda et al. (2011) in this section.** We agree the Sakaeda et al study is valuable, as there the semi-direct and direct effects are separated (unlike in the study of Lu et al, which we already cite). However, there is no separation between coastal and remote areas as in Lu et al., and the non-monotonic nature of the changes to the temperature profile we see (Figure 13) preclude a straightforward comparison of the LTS between simulations with and without smoke as Sakaeda et al do. We added *The study of Sakaeda et al (2011) does separate the semi-direct and direct effects (but does not consider the indirect effect). They find a small negative semi-direct effect, averaging $-2.6\,\mathrm{Wm}^{-2}$ at the top of the atmosphere, and a small positive direct effect, averaging $+0.9\,\mathrm{Wm}^{-2}$. This seems broadly consistent with Lu et al (2018). We note that both Lu et al (2018) and Sakaeda et al (2011) also average over pristine periods without much smoke aerosol, while we quote results only in a very polluted period.*

We were also inspired by this comment to compare the direct radiative effect more quantitatively to the radiative forcing efficiency found by Chand et al (2009), as Sakaeda et al do, and we modified the text in the previous subsection appropriately. We included a new supplementary figure showing the radiative forcing efficiency as a function of the cloud fraction.

**2 Reviewer 2**

1. **Page 5: if winds are being nudged, would that not diminish any dynamic response of large-scale vertical motion to a heating perturbation?** The nudging of the horizontal winds probably does diminish dynamic responses to heating, yes. Without any nudging, on the other hand, the comparison with observations gets much more difficult. The alternative

is the approach of Lu et al (2018), who run simulations in forecast mode, reinitialised every three days. However, like nudging, this may also lead to a reduction in the overall simulated effect of the aerosols. To study this further we are planning a follow-up paper where we will compare nudged simulations to simulations in forecast-mode as done by Lu et al (2018). For now, we added a note to remind the readers of this possible source of bias in our conclusions: *Nudging our global model to horizontal winds may also artificially affect the vertical transport of, and dynamical responses to, the smoke aerosol.*

We thank the reviewer for the various comments on model configuration. For the current manucript, we prefer not to re-run the simulations with a new tuning, but we try to make clear in the manuscript that the choices are sometimes quite arbitrary and should be updated in future. We will advertise our manuscript and your helpful comments to climate modellers in the UK research community.

2. **Surprising that changing $\sigma_w$ from eqn. 1 to just fixing it at $0.12\,\mathrm{m/s}$ would have much of an effect in increasing CDNC, given that eqn. 1 already includes a floor of $0.10\,\mathrm{m/s}$. Also, why not just increase the floor to $0.12\,\mathrm{m/s}$ in the original expression?** We initially fixed the vertical velocity in the activation scheme, instead to using a function of TKE, in order to make the model more similar to the cloud microphysics scheme CASIM that we are developing in parallel to this work. It made the vertical velocity easier to tune. With hindsight, we should indeed have just tuned the floor instead. All of these options are rather artificial, and we hope to study this more properly in future work, perhaps drawing on Malavelle et al, GRL 2014, which provides a better motivated approach to vertical velocity in low resolution models.

3. **The kappa value for sulphate seems quite large. Typical values for ammonium bisulphate are more like 0.6, maybe 0.7 at the extreme (cf. Petters and Kreidenweiss 2007). Using a value of effectively unity seems worth an explanation.** The kappa value for sulphate is carried over from the version of the Unified Model we inherited, and dates back to the time when so-called van t'Hoff factors were more commonly used than kappa values. It is based on the assumption that 3 ions per sulphuric acid molecule are formed on activation, which is indeed wrong, it is more like 2 ions per molecule. For our study, since the vertical velocity is tuned, the most important thing is the ratio of the hygroscopicity of sulphate to that of OC, which determines the ratio of CDNC in the pristine and polluted periods. The absolute values are not so important. If we had used a lower hygroscopicity for sulphate, we could also have used a lower hygroscopicity for OC, which would also be more realistic. When we improve the treatment of activation in our model, we will be able to make this more physical. At this point we will probably also need to think more carefully in general about the aerosol composition, for example including biomass burning sulphate emissions. We think it is fairly clear from the text as it stands that additional work is needed here, but we now explicitly point out that the hygroscopicity for sulphate is unrealistically high, as well as that of OC.

4. **What is basis for fraction of each grid box that is raining being 0.3 on the 65 km mesh?** The global rain fraction of 0.3 follows the assumption by Mann et al (2010). We added the citation. We don't know whether there are any calculations to justify the number.

5. **Interquartile range is a scalar, and what is shaded in fig. 3 is the inner half of the distribution.** We fixed the description of the shaded region in Figure 3.

6. **Might point out in the last paragraph of section 6.1 that while the moisture aloft is indeed associated with BBA aloft over the region, they are associated with a com- mon factor, namely outflow from the deep continental boundary layer, and surely the moisture would be there even in the absence of aerosol sources.** We agree we should emphasise the origin of the moisture, and we added a comment to clarify the origin of the water vapor, *Like the smoke, the water vapor originates from the continental boundary layer. This is not to say that it is co-emitted with the biomass burning emissions: we do not simulate such emissions of water vapour in our model.*

7. **The generally good agreement between aerosol index AI and aerosol optical depth AOD described in the first full paragraph of page 19 does not get so much support from fig. 8, which instead shows that 2 of the 5 so-called clean days have quite large values of AI but not AOD. Not sure I'd characterize the relationship as generally good and wonder how robustly they are correlated.** Indeed, while there is some correlation between AI and AOD, maybe 'generally good' was a bit optimistic. AOD doubles between 4 and 7 August while AI is already high on 4 August and only increases by about 50% between 4 and 7. We plotted the spatial variability of AI and of simulated and observed AOD in a supplementary figure. It shows how we sampled the observed AI and AOD, which, due to the nature of the satellite retrievals, is not always uniformly across our domain. It also shows more clearly that AI increases before AOD, as the reviewer suggests. We think the high altitude of the aerosol layer (3 km on 2nd August, from our CALIPSO feature mask, figure 4) means AI may be more sensitive to it than AOD. AOD should be roughly independent of height, while the reflectivity residue, from which AI is calculated, increases linearly with height (for reasonably low single scattering albedo, as per Figure 1 of Herman et al, JGR 1997). We removed the comment about generally good correlation and replaced it with *Despite the better correlation of aerosol number concentration expected with AI than AOD, the AI increases before the AOD while the CCN at Ascension Island increases after. This is likely due to the vertical mixing of the aerosol layer. The layer is initially at high altitude, which leads to a higher AI than lower-lying aerosol layers would (Herman et al, 1997) but does not affect AOD. Then, as the layer thickens, the AOD increases more quickly than the AI. The spatial patterns of AI, retrieved AOD and simulated AOD are shown in Supplementary Figure S8.*

8. **MODIS platforms pass overhead not far from local noon. Do the comparisons with model result in fig. 10 use the same, quite limited sampling of the diurnal cycle?** Yes, the sampling of the diurnal cycle is rather limited in the end. For liquid water path we are able to add AMSR but for CDNC we just use MODIS. We explain in Section 4 and the SI that we also tried to use SEVIRI data. As the community of satellite specialists looking at SEVIRI data is now growing thanks to CLARIFY and ORACLES, there are good prospects for exploiting this data more fully in subsequent modelling studies.

9. **The positive LW semi-direct forcing is confusing, given that lower cloud tops for polluted clouds should emit more LW. Less emission from even warmer surface through more overcast deck, perhaps?** The positive long-wave

semi-direct forcing happens mainly on 6, 7 and 8 August, except for a brief reversal on the evening of the 6th. On these days the difference in cloud fraction between simulations with and without smoke is highest, at around 20%. We agree with the assessment of the reviewer that the extra long-wave that makes it through the clouds from the sea surface means the overall long-wave cooling is higher in the simulation without smoke. This presumably more than cancels out the effect of the clouds being higher and thinner without smoke, and emitting less longwave. The difference between outgoing longwave from clouds and the surface is about $15\,\mathrm{Wm}^{-2}$, for an average outgoing LW around $290\,\mathrm{Wm}^{-2}$. If the cloud fraction changes from 0.5 to 0.7 due to aerosol absorption, this would change the outgoing LW by $3\,\mathrm{Wm}^{-2}$. This number is quite close to the observed semi-direct effect, so the explanation is not unreasonable. We replaced our previous suggestion of an aerosol effect with *The semi-direct effect may well be due to a reduction in the long-wave radiation from the surface reaching the top of the atmosphere, as the cloud fraction increases when smoke aerosol is present.*

10. **The entrainment rates discussed on page 33 seem higher than expected, with values of around 2 to 5 cm/s. Not hard to find literature reporting typical rates for nocturnal stratocumulus 5 if not 10 times smaller, even smaller during the day. Is mean entrainment rate computed from the mean of column-wise $\mathrm{d}(z_i)/\mathrm{d}t - w(z_i)$? The plausible simulation of PBL depth does not obviously square with the reported rates.** The entrainment rates we present are domain means. In cloud-free or cumulus areas of our domain, we find that they are substantially higher than in stratocumulus areas. We now show plots of the spatial distribution of entrainment rates together with the spatial distribution of cloud fraction in our supplementary materials.

We are able to dig a bit deeper into this. We have a 'boundary layer type' diagnostic in our model, which allows us to stratify the entrainment rate by the boundary layer type. Boundary layer type 3 is a well-mixed boundary layer, 4 is decoupled stratocumulus not over cumulus, 5 is decoupled stratocumulus over cumulus, and 6 is for cumulus. As shown in the figures below, type 5 is the most common, followed by type 6 (especially in daytime) and then types 3 and 4. The entrainment rates for cumulus (type 6) are around $8\,\mathrm{cm/s}$, while for our most common stratocumulus-topped boundary layer type (type 5) the entrainment rates are down at or below around $1\,\mathrm{cm/s}$. There is the expected diurnal cycle, though it is quite weak.

We added to the text *In our simulation the entrainment rates for cumulus or clear skies substantially exceed those for stratocumulus, which are around $1\,\mathrm{cm s}^{-1}$ in the regional model and $0.5\,\mathrm{cm s}^{-1}$ in the global model.*

**Typesetting issues**

1. **Line 6 of page 3 should use \citet instead of \citep.**

2. **Year for Gunnar et al. citation is missing on line 12 of page 6.**

3. **The citation D. Holdridge...2016a appears in multiple places but is missing from the references.**

We fixed the three typesetting issues, thank you for spotting these errors.

[Figure]

**Figure 1.** Left, frequency of occurrence of the different boundary layer type variables in our regional model domain, and right, entrainment rates of the different boundary layer type in our regional and global models. The fluctuations in the global model reflect the small number of grid boxes with a given boundary layer type in the regional model domain.

[revised manuscript text omitted]

**Supplementary 1: satellite intercomparison**

The LWP from SEVIRI is calculated using the formula

$$\frac{2}{3}\tau_{vis}r_e\rho_l \tag{S1}$$

where the cloud optical thickness $\tau_{vis}$ is retrieved at $1.6\,\mu m$; $r_e$ is the effective radius, also retrieved at $1.6\,\mu m$, and $\rho_l$ is the density of liquid water (Roebeling et al., 2008a). For MODIS the same equation is used except for multiplication by a factor $2/Q_e$, where $Q_e \sim 2$ is the extinction efficiency (King et al., 2013).

The cloud droplet number $N_d$ is calculated from MODIS data using the equation

$$N_d = \frac{\sqrt{5}}{2k\pi}\frac{(f*C_w*\tau_{vis})^{1/2}}{Q\rho_l r_e^{5/2}} \tag{S2}$$

where $k = 0.8$, $Q = 2$, $f = 0.7$, and $C_w$ is the rate of increase of liquid water content with height (Boers et al., 2006; Grosvenor and Wood, 2014),

$$C_w = \frac{C_p}{L_v}(\Gamma_d - \Gamma_m)\rho_a \tag{S3}$$

. In this last equation, $\Gamma_d$ and $\Gamma_m$ are the dry and moist adiabatic lapse rates, $C_p$ the heat capacity of dry air, $\rho_a$ its density, and $L_v$ the latent heat of condensation of water. The SEVIRI calculation is similar (Roebeling et al., 2008b).

**Table S1.** Comparison of optical retrievals: normalized mean bias (b) and Pearson correlation coefficient (r) of SEVIRI relative to MODIS AQUA. The data are all regridded to the coarser AMSR resolution, to avoid artefacts from slight mis-alignment of the model grids.

| Day | $R_e$ b | $R_e$ r | COT b | COT r | $N_d$ b | $N_d$ r |
|-----|---------|---------|-------|-------|---------|---------|
| 1 | -0.20 | 0.53 | -0.29 | 0.59 | -0.34 | 0.64 |
| 2 | -0.24 | 0.68 | -0.20 | 0.93 | -0.28 | 0.63 |
| 3 | -0.22 | 0.71 | -0.18 | 0.82 | -0.31 | 0.83 |
| 4 | -0.40 | 0.71 | -0.34 | 0.90 | +0.08 | 0.77 |
| 5 | -0.34 | 0.64 | -0.16 | 0.79 | +0.41 | 0.53 |
| 6 | -0.37 | 0.83 | -0.19 | 0.93 | +0.33 | 0.54 |
| 7 | -0.25 | 0.59 | -0.15 | 0.95 | -0.07 | 0.55 |
| 8 | -0.12 | 0.24 | +0.01 | 0.66 | -0.46 | 0.35 |
| 9 | -0.13 | 0.60 | -0.29 | 0.86 | -0.47 | 0.71 |
| 10 | -0.06 | 0.82 | -0.28 | 0.68 | -0.47 | 0.77 |

**Table S2.** Comparison of liquid water path retrievals: normalized mean bias (b) and Pearson correlation coefficient (r) of MODIS AQUA and SEVIRI relative to AMSR, on the AMSR resolution.

| Day | AQUA b | AQUA r | SEV b | SEV r |
|-----|--------|--------|-------|-------|
| 1 | -0.23 | 0.48 | -0.37 | 0.16 |
| 2 | -0.12 | 0.69 | -0.32 | 0.63 |
| 3 | -0.31 | 0.64 | -0.41 | 0.58 |
| 4 | -0.11 | 0.61 | -0.46 | 0.48 |
| 5 | -0.15 | 0.49 | -0.42 | 0.43 |
| 6 | -0.02 | 0.77 | -0.48 | 0.74 |
| 7 | -0.14 | 0.65 | -0.37 | 0.65 |
| 8 | -0.23 | 0.49 | -0.21 | 0.38 |
| 9 | -0.15 | 0.67 | -0.21 | 0.61 |
| 10 | -0.12 | 0.32 | -0.11 | 0.47 |

[Figure]

**Figure S1.** CALIOP vertical feature mask and aerosol optical depth (filtered such that the AOD uncertainty is below 0.05) for 7 August (the left plot is identical to the bottom subplot of Figure 5).

[Figure]

**Figure S2.** MODIS (left) and SEVIRI (centre) cloud droplet effective radius (at $3.7\,\mu\mathrm{m}$, cloud optical thickness, and cloud top temperature, on 2nd August 2016, and the correlation of coarse-grained values (in AMSR $14\,\mathrm{km}$ grid boxes), right. Liquid water paths below $10\,\mathrm{g}\,\mathrm{m}^{-2}$ are screened out, for clarity. Summary statistics for all ten days of the study period are given in Table S1 below.

[Figure]

**Figure S3.** MODIS (left) and SEVIRI (centre) cloud droplet effective radius, cloud optical thickness, and cloud top temperature, on 7th August 2016, and the correlation of coarse-grained values (in AMSR 14 km grid boxes), right. Liquid water paths below $10\,\mathrm{g\,m^{-2}}$ are screened out, for clarity.

[Figure]

**Figure S4.** MODIS cloud droplet number concentration and liquid water path (left) on 2nd and 7th August 2016, and, right, SEVIRI cloud droplet number concentration and AMSR liquid water path.

**Supplementary 2: additional simulation output**

[Figure]

**Figure S5.** Black carbon mass loading (left) and relative humidity (right) at the specified altitudes, 150 (top) and 180 (bottom) hours into the simulation.

[Figure]

**Figure S6.** Temperature gradient across the inversion at Ascension Island in the radiosonde observations from the ARM site (D. Holdridge, J. Kyrouac and R. Coulter, 2016b), plotted against the values from the model. The difference between the maximum and minimum temperatures around the inversion is divided by the difference in height between these extrema.

[Figure]

**Figure S7.** Observed and modelled wind speed profiles at Ascension Island, from soundings on 1,3,5,7,9 August (D. Holdridge, J. Kyrouac and R. Coulter, 2016b). Wind speed is plotted on the $x$ axis and altitude in metres on the $y$ axis of each sub-plot. The soundings are compared to the regional simulation, marked as 'Sim U' and 'Sim V'.

[Figure]

**Figure S8.** Aerosol index (top) from OMPS, aerosol optical depth (AOD) from MODIS AQUA (middle) and AOD from the regional model (bottom) on 1,3,5,7,9 August. The MODIS AOD is filtered to exclude regions above cloud, before being regridded to the model grid. The regridding algorithm excludes cloudy regions, white on the plots, where valid pixels are greater than $0.25°$ (28 km) apart (we assume that AOD is slowly varying on lengthscales smaller than this, so a retrieval will be representative of neighbouring areas where cloud is beneath the aerosol). The aerosol index is only retrieved from OMPS if it is greater than around 0.5.

[Figure]

**Figure S9.** Cloud fractions from the Unified Model cloud schemes at 0130 UTC(top) and 1330 UTC (bottom) on 7 August. . The left plot shows the regional model, then the regional model regridded onto the global grid, then the global model, and finally the histogram of cloud fractions in model grid boxes. In all plots the maximum cloud fraction in an atmospheric column is shown.

[Figure]

**Figure S10.** Entrainment rates at the top of the boundary layer at 0130 UTC(top) and 1330 UTC (bottom) on 7 August. The left plot shows the regional model, then the regional model regridded onto the global grid, then the global model, and finally the histogram of entrainment rates in model grid boxes.

[Figure]

**Figure S11.** Shortwave, longwave and net domain mean heating rates during the ten-day model simulation in the regional model. The dotted vertical lines indicate midnight, local time. Cloud base and cloud top are also marked.

[Figure]

**Figure S12.** Domain-mean vertical profiles of potential temperature Θ and liquid water content (LWC) with and without biomass burning aerosol on odd numbered days at the times of day marked, from the global model.

[Figure]

**Figure S13.** Outgoing shortwave flux at top of atmosphere for the baseline model and a model version with aerosol absorption set to zero, for the regional domain in the global model on the left and the regional model on the right. CERES domain-mean observations are shown in the same figure in red crosses (some of these may be biased due to the gaps between swaths). The changes to CDNC and liquid water path across the 10 days are also shown.

[Figure]

**Figure S14.** Decomposition of the short-wave and long-wave radiative effects of biomass burning aerosol using simulations with the microphysical parameterisation of Kogan (2013) instead of the default, from Khairoutdinov and Kogan (2000).

[Figure]

**Figure S15.** Cloud fraction with and without biomass burning emissions in the regional model, and in the same domain in the global model. The simulations without aerosol absorption are also shown.

[Figure]

**Figure S16.** Radiative forcing efficiency (Chand et al., 2009), defined as the direct effect divided by the aerosol optical depth, as a function of cloud fraction $f_c$ in the clean and polluted periods of our regional model simulation. The linear fit shown for the polluted period has equation $125.4 f_c - 53.7$.